# HIERPROMPTLM: HIERARCHICAL PROMPT LANGUAGE MODEL FOR HETEROGENEOUS TEXT-RICH NETWORKS

## ABSTRACT

Representation learning on heterogeneous text-rich networks (HTRNs), consisting of multiple types of nodes and edges with each node associated with text data, is essential for various real-world applications. Given the success of pretrained language models (PLMs) in processing text data, recent efforts have integrated PLMs into HTRN representation learning, typically handling textual and structural information separately with PLMs and heterogeneous graph neural networks (HGNNs), respectively. However, this separation fails to capture critical interactions between these two types of data, and necessitates alignment between distinct embedding spaces, which is often challenging. To address this, we propose **HierPromptLM**, a novel pure PLM-based framework that models text data and heterogeneous structures without separate processing. First, we develop *Hierarchical Prompt* that employs prompt learning to integrate text data and structures at both node and edge levels, within a unified textual space. Built on this, two innovative *HTRN-tailored pretraining* tasks are introduced to fine-tune PLMs, emphasizing the heterogeneity and interactions between these two types of data. Experiments on HTRN datasets demonstrate HierPromptLM outperforms state-of-the-art methods, achieving significant improvements of up to 7.15% on node classification, 9.79% on link prediction, and 2.88% on graph classification. The codes are in
`https://anonymous.4open.science/r/HierPromptLM-code`.

## 1 INTRODUCTION

Heterogeneous text-rich networks (HTRNs) (Shi et al., 2019; Zhang et al., 2019a; Jin et al., 2023b), composed of multiple types of nodes and edges where each node is associated with textual information, have been widely used to model various real-world scenarios, such as social media (El-Kishky et al., 2022), academic networks (Tang et al., 2008) and product networks (Dong et al., 2020). For example, the academic data shown in Figure 1 can be represented as an HTRN, comprising three types of nodes (*i.e.,* paper, author, venue) and three types of

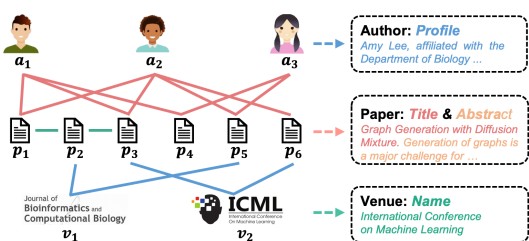

Figure 1: Illustration of an HTRN.

edges (*i.e.,* author-write-paper, venue-publish-paper, paper-cite-paper). Here, papers are associated with abundant textual information, including titles and abstracts, which are commonly named text-rich nodes in prior works (Jin et al., 2023b; Zou et al., 2023). Venues, on the other hand, contain only name information, typically referred to as textless nodes (Zou et al., 2023). Due to their important roles, representation learning on HTRNs has gained increasing attention in recent years as a powerful tool to embed nodes and/or edges into meaningful representations, which facilitates various downstream tasks.

The most intrinsic aspect of HTRNs is their heterogeneous structures, represented by diverse nodes with their complex connections. Heterogeneous Graph Neural Networks (HGNNs), designed for heterogeneous graph data, have been extensively utilized in the literature (Yang et al., 2021b; 2022) to capture such structures. For example, HAN (Wang et al., 2019) proposes a hierarchical neighbor aggregation strategy that adopts both node-level and meta-path-level attentions to aggregate

information from diverse types of nodes. Built on this, MAGNN (Fu et al., 2020) further considers intermediate nodes along meta-paths, leveraging both intra-meta-path and inter-meta-path information for high-order neighbor aggregations. However, these studies primarily focus on modeling structural information, neglecting additional textual information unique to HTRNs, which greatly affects their effectiveness for HTRN representation learning (Jin et al., 2023b).

To deal with the textual information in HTRNs, recent models have further extended HGNNs by integrating additional pretrained language models (PLMs) (Kenton & Toutanova, 2019; Chi et al., 2021) component, drawing inspiration from the significant success of PLMs in natural language processing (NLP) (Radford et al., 2019; Chi et al., 2021). For example, Heterformer (Jin et al., 2023b) learns a target node's representation by aligning the structural embedding generated by HGNNs from its one-hop neighbors with the textual embedding from auxiliary PLMs. THLM (Zou et al., 2023) leverages PLMs to enrich node representations on HTRNs, making them text-aware by aligning the embedding spaces of HGNNs and PLMs through a binary matching task.

Despite recent improvements, these methods process text data and heterogeneous structures separately, overlooking their crucial interactions. Specifically, consider a typical node classification scenario for an author node, illustrated in Figure 1, where author $a_3$ frequently collaborates with other researchers in machine learning area (*i.e.,* $a_1$ and $a_2$), while $a_3$'s own textual profiling indicates a primary focus on biology. By explicitly integrating the interplay between text data and structures, it becomes clear that the author applies machine learning techniques to biological problems, correctly identifying $a_3$ as working in computational biology field. Additionally, existing approaches that rely on *separate processing* often necessitate extra alignment, which is challenging due to the fundamental differences in the embedding spaces generated by PLMs and HGNNs. For instance, in the HTRN depicted in Figure 1, the biology paper $p_2$ has several machine learning papers (*i.e.,* $p_1$ and $p_3$) as neighbors resulting from citation relations. Enforcing alignment between this paper's textual embedding, which indicates a focus on biology, and its structural embedding, which aggregates information from machine learning-focused neighbors, may introduce misleading signals to the model learning.

To overcome these shortcomings associated with *separate processing*, we adopt a fundamentally different perspective and propose to jointly incorporate both textual and structural information within HTRNs in a unified representation space for the first time. To achieve this, we introduce a pure PLM-based framework, named **HierPromptLM** (**Hier**archical **Prompt L**anguage **M**odel), given that PLMs have learned massive context-aware knowledge and demonstrated promising results in encoding either text (Kenton & Toutanova, 2019) or graph data (Ye et al., 2024). Our framework features the specialized *Hierarchical Prompt*, which leverages prompt learning to capture heterogeneous structures, seamlessly integrating them with text data into a unified textual space, where interactions can be naturally captured in PLM without extra alignment. Unlike existing methods (Guo et al., 2023; Tang et al., 2024) that primarily focus on homogeneous graphs, *Hierarchical Prompt* introduces an automated textualization mechanism specifically designed for heterogeneous graphs and emphasizes the inherent heterogeneity at node and edge levels, which leads to two corresponding designs: graph-aware prompt and relation-aware prompt. We explain their intuitions and key ideas below.

**(1) Graph-aware prompt** extracts and composes meaningful text sequences from a target node's subgraph context, enriching its representation from a structural perspective in addition to its own text data. Due to inherent heterogeneity in a node's subgraph context, we decompose it into distinct meta-path-based subgraphs with pre-defined meta-paths, as shown in Figure 2(a), inspired by their effectiveness in capturing heterogeneous structures (Wang et al., 2019; Hu et al., 2020; Fu et al., 2020). These subgraphs enhance the target node's representation through structural augmentation, enabling to capture a broader context while distinguishing the fine-grained semantics of different neighbors via various meta-paths. By designing a meta-path-based subgraph program function to textualize these subgraphs into graph summaries and combining them with the node's own text data, we create a unified text sequence to jointly handle both types of information. However, such straightforward approach often generates lengthy inputs, which poses challenges for PLMs due to token length limits (Bi et al., 2024). To address this, we employ a frozen PLM to distill subgraph summaries into graph tokens before concatenation, serving as specialized soft prompts and resulting in a hierarchical graph-aware prompt for each target node. **(2) Relation-aware prompt** captures heterogeneous edge-level structures in HTRNs. To explicitly capture the heterogeneity of relations, we introduce a learnable relation token, another special soft prompt designed to capture diverse interactions between different node pairs. This token is positioned between the graph-aware prompts of a pair of nodes, forming the relation-aware prompt. Through such prompt design, we reformulate

both heterogeneous structures and text data unique to HTRNs into a unified text sequence, marking the first attempt in this field to the best of our knowledge.

The obtained relation-aware prompt is fed into a tunable PLM for representation learning. To further enhance the effectiveness of representation learning on HTRNs, we introduce two *HTRN-tailored pretraining* tasks, *i.e.,* **H**eterogeneous **G**raph-**a**ware **M**asked **L**anguage **M**odeling (HGA-MLM) and **H**eterogeneous **G**raph-**a**ware **N**ext **S**entence **P**rediction (HGA-NSP), which jointly capture the heterogeneity and text-rich characteristics underexplored by existing HTRN-based methods. Once trained, our framework produces versatile, unsupervised representations for both nodes and edges, facilitating various downstream applications. Our main contributions are summarized below.

- For the first time, we propose a novel pure PLM-based method, HierPromptLM, for HTRN representation learning, which features in an innovative hierarchical prompt framework that seamlessly integrates text data with heterogeneous node-level and edge-level structures, along with their interactions, into a unified textual space.
- To the best of our knowledge, we are the first to propose two new *HTRN-tailored pretraining* tasks for PLMs, enhancing representation learning on HTRNs.
- Experiments on three datasets demonstrate HierPromptLM significantly outperforms 15 baselines on two typical downstream tasks, with notable improvements of up to 7.15% on node classification, 9.79% on link prediction, and 2.88% on graph classification compared to state-of-the-art methods.

## 2 RELATED WORK

### 2.1 PRETRAINED LANGUAGE MODELS

The primary goal of PLMs (Kenton & Toutanova, 2019; Raffel et al., 2020; Chen et al., 2024) is to learn general text representations from large-scale, unlabeled corpora through pretraining tasks, which can be applied to various downstream NLP tasks (Liu, 2019a; Meng et al., 2022). Driven by the fact that a word's meaning can vary depending on its context, models such as BERT (Kenton & Toutanova, 2019) and T5 (Raffel et al., 2020), are introduced to generate contextualized token representations that adapt to their surrounding text. However, these models mainly focus on text encoding only. Recently, several studies (Chien et al., 2021; Levine et al., 2021; Yasunaga et al., 2022) have been proposed to further enhance the capture of connections between different text. Despite these advancements, none of them are specifically designed for HTRNs, which requires to consider both text data and the heterogeneity of relations. To address this gap, we propose two newly designed HTRN-tailored pretraining tasks, which jointly captures the inherent heterogeneity and text-rich characteristics of HTRNs.

### 2.2 TEXT-RICH NETWORK MINING

Text-rich networks (TRNs) (Jin et al., 2023a; Yu et al., 2023), where nodes contain textual information, are widely used in real-world applications. With the success of PLMs in handling text data, existing methods (Yang et al., 2021a; Zhu et al., 2021; Li et al., 2021) combine both GNNs and PLMs to explore the textual and structural information within TRNs, but they often assume homogeneous networks, which is impractical in real life. Thus, recent research has focused on HTRN representation learning. For example, Heterformer (Jin et al., 2023b) combines structural embeddings from its direct neighbors generated by HGNNs with textual embeddings from auxiliary PLMs. To capture higher-order structures, THLM (Zou et al., 2023) aligns the embedding spaces of HGNNs and PLMs via a context graph prediction task. However, these methods treat structural and textual information separately, failing to fully capture their interactions and requiring additional alignment due to the differences between the embedding spaces of PLM and HGNN. Different from these works, we adopt a fundamentally different perspective by proposing the first pure PLM-based framework that jointly models text data and heterogeneous structures in a unified representation space.

## 3 PRELIMINARIES

**Definition 3.1. Heterogeneous Text-Rich Networks.** A heterogeneous text-rich network (HTRN) is defined as $\mathcal{G} = (\mathcal{V}, \mathcal{E}, \mathcal{C}, \mathcal{R}, \mathcal{T})$, where $\mathcal{V}, \mathcal{E}, \mathcal{C}, \mathcal{R}, \mathcal{T}$ represent the set of nodes, edges, node types,

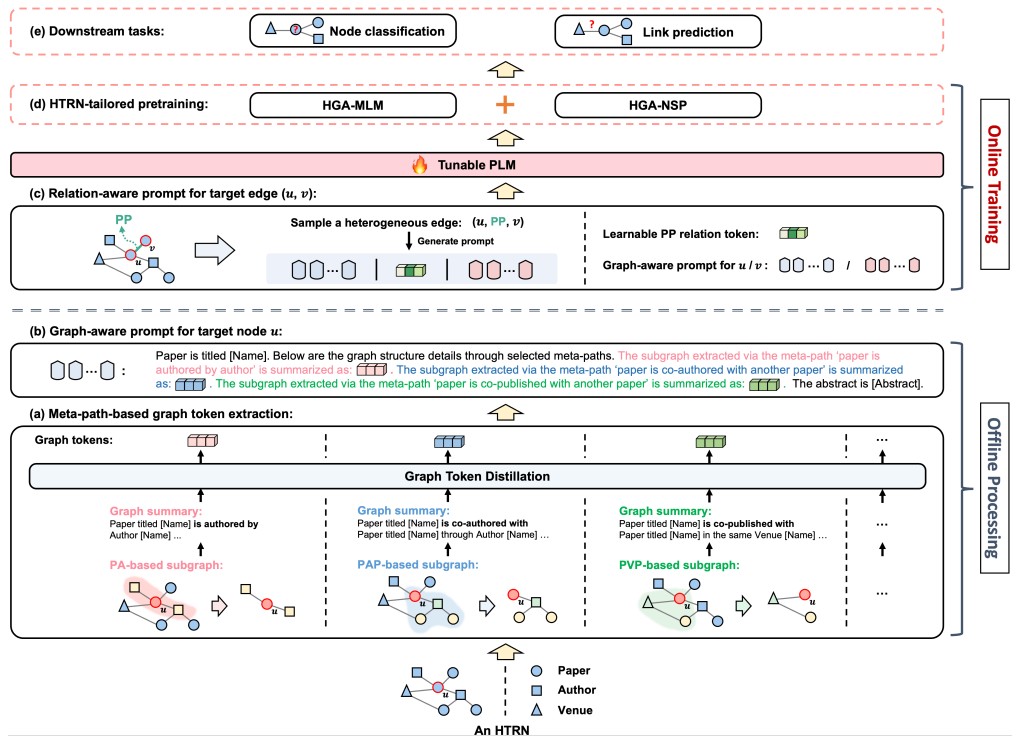

Figure 2: The framework of HierPromptLM, where graph-aware prompt generation can be conducted offline to enhance efficiency. (a) Extract meta-path-based subgraphs, create corresponding meaningful textual summaries, and distill into graph tokens using a frozen PLM. (b) Generate a graph-aware prompt by integrating the node's own text and graph tokens with their descriptions. (c) Generate a relation-aware prompt by combining each node's graph-aware prompt with a learnable relation token. (d) Fine-tune a PLM with the relation-aware prompt using two specialized pretraining tasks.

relation types and textual descriptions, respectively. Each node is associated with a node type $c \in \mathcal{C}$ and the textual information $t \in \mathcal{T}$, and each edge is characterized by a relation type $r \in \mathcal{R}$. Note that an HTRN should satisfy the condition $|\mathcal{C}| + |\mathcal{R}| > 2$.

**Definition 3.2. HTRN Representation Learning Problem**. Given an HTRN, our goal is to fine-tune a PLM $f_\Theta$ parameterized by $\Theta$, which effectively captures the textual and heterogeneous structural information of the HTRN and can be utilized to generate both node and edge representations to facilitate various downstream tasks. Specifically, for a node $u$ in the HTRN, we aim to learn its embedding $z_u = f_\Theta(u)$, where $z_u \in \mathbb{R}^d$, and $d$ denotes the embedding dimension. Similarly, given a pair of nodes $u$ and $v$ connected by relation $r$, which forms an edge $e = (u, r, v)$, we aim to learn the edge embedding $z_e = f_\Theta(e)$ where $z_e \in \mathbb{R}^d$.

## 4 METHODOLOGY

Figure 2 illustrates the framework of our HierPromptLM, which comprises two components: (1) *Hierarchical Prompt*: creates a graph-aware prompt by augmenting the target node's representation with its surrounding context to capture node-level heterogeneity, followed by a relation-aware prompt to further capture edge-level heterogeneity. This reformulates both text and heterogeneous structures into a PLM-interpretable text sequence, capturing critical interactions without alignment. (2) *HTRN-tailored Pretraining*: introduces HGA-NSP and HGA-MLM, two new tasks for fine-tuning PLMs via relation-aware prompts, highlighting the heterogeneity and rich textual context within HTRNs.

### 4.1 GRAPH-AWARE PROMPT

To capture each node's heterogeneous structures, we design a graph-aware prompt that, for the first time, leverages meta-path-based subgraphs surrounding the target node for structural augmentation.

### 4.1.1 Meta-path-based subgraph

Inspired by the effectiveness of meta-paths in capturing local context (Sun et al., 2011; Wang et al., 2019), we introduce them to model the heterogeneous structures of each target node. Specifically, in HTRNs, nodes are connected through various semantic paths, known as meta-paths.

**Definition 4.1. Meta-path.** A meta-path $m$ is a path $c_1 \xrightarrow{r_1} c_2 \xrightarrow{r_2} \ldots \xrightarrow{r_k} c_{k+1}$ (abbreviated as $c_1 c_2 \ldots c_{k+1}$), which describes a composite relation $R$ connecting two endpoint nodes through a series of intermediate nodes and relations, where $c_i$ denotes node types and $r_i$ denotes relation types.

Built upon the concept of meta-path, we first introduce a new structure named the meta-path-based subgraph, which is formally defined as follows.

**Definition 4.2. Meta-path-based Subgraph**. Given a node $u$ and a meta-path $m$ in HTRNs, the meta-path-based subgraph $G_m(u)$ consists of node $u$, its meta-path-based neighbors (which refer to nodes connected to $u$ via meta-path $m$), and all intermediate nodes along the path.

Taking Figure 2 as an example, for a target node $u$ with a predefined meta-path $PAP$, the extracted $PAP$-based subgraph of $u$ (shaded in blue) includes the target node (red), its meta-path-based neighbors (yellow), and intermediate nodes (green). This structure captures multi-level information, each with unique meanings: the target node at distance 0 reflects its own characteristics, intermediate nodes at distance 1 indicate authorship, and meta-path-based neighbors at distance 2 signify co-authorship. By incorporating such neighbors at multiple levels, the model gains a comprehensive view of the target node, from its intrinsic characteristics to broader structural context within HTRNs.

Different meta-paths divide a target node's local structure into distinct subgraphs with different semantic aspects. For example, the $PA$-based subgraph (shaded in red) reflects collaboration, while the $PVP$-based subgraph (shaded in green) highlights shared research topics. Together, all these subgraphs capture the broader context and enhance the target node's representation within HTRNs.

**Meta-path-based Graph Summarization.** To integrate these subgraphs into a textual space, we design a subgraph program function $\mathcal{P}(\cdot)$ to generate textual summaries by combining the textual information of nodes in the subgraph with meta-path's semantics. For instance, in Figure 2(a), consider a target node $u$ with information $\Phi(u) = \{$Meta-path: $PA$; $PA$ semantics: *"is authored by"*; $PA$-based subgraph $G_{PA}(u)$: $\{u, Author1, Author2\}\}$, we generate the summary as $\mathcal{P}(u, G_{PA}(u)) =$ <Paper titled [Name of $u$] is authored by Author [Name of Author1], Author [Name of Author2]>.

Note that our method is easily extendable to accommodate more complex meta-paths and efficiently generates meta-path-based graph summaries. This process is highly parallelizable, which enables simultaneous generation of numerous subgraph summaries during offline preprocessing.

### 4.1.2 Meta-path-based graph tokens

Directly concatenating various meta-path-based subgraph summaries with the target node's text often results in lengthy inputs, which may exceed PLMs' token limits (Bi et al., 2024). To mitigate this, we design a novel hierarchical structure to first distill these summaries into concise semantic tokens (called graph tokens) before their integration, using a frozen PLM without additional training. This process is executed efficiently offline and supports parallel processing.

### 4.1.3 Prompt generation

Generated graph tokens, acting as specialized soft prompts, are then merged with the target node's own text via a graph-aware program function $\mathcal{F}(\cdot)$ to construct a graph-aware prompt. For example, for a target node $u$ with information $\phi(u) = \{PA$-based graph token: $\mathcal{T}_G(u, PA)$, Token descriptions: *"The subgraph extracted via the meta-path 'paper is authored by author' is summarized as:"*$\}$, the graph-aware prompt is: $\mathcal{F}(u) =$ <Paper is named [Name of $u$]. The subgraph extracted via the meta-path 'paper is authored by author' is summarized as: $[\mathcal{T}_G(u, PA)]$>. Additional node-specific information, such as abstracts can be appended, *e.g.,* "The abstract is: [Abstract of $u$]". Formally, the graph-aware prompt for the target node is defined as: $\mathcal{F}(u) =$ <[Text] [Token Descriptions] [Graph Token]>, where "Text" refers to the target node's textual information, "Graph Token" represents distilled graph tokens, and "Token Descriptions" provides their semantic descriptions. This structured approach offers a flexible yet consistent framework for various node types, seamlessly integrating

rich text and heterogeneous structures into a unified textual space. Notably, the graph-aware prompts, involving precomputed graph tokens, can be efficiently generated offline in parallel.

## 4.2 RELATION-AWARE PROMPT

Beyond node-level modeling with the graph-aware prompt, it is crucial to further capture heterogeneous structures more comprehensively from a complementary edge-level perspective, as edges in HTRNs represent critical relations between various types of nodes, such as authorship and citation, essential for understanding node interactions within HTRNs.

To address such heterogeneous edge-level information and effectively integrate it with rich textual information into a unified space, we propose the relation-aware prompt. This module introduces a learnable relation token, strategically placed between the graph-aware prompts of connected nodes to form a comprehensive relation-aware prompt. Specifically, for a node pair $(u, v)$ connected by a relation $r \in \mathcal{R}$, the relation-aware prompt $\mathcal{H}(u, r, v)$ is defined as: $\mathcal{H}(u, r, v) = <[\mathcal{F}(u)]$ [Relation Token of $r$] $[\mathcal{F}(v)]>$, where $\mathcal{F}(u)$ and $\mathcal{F}(v)$ denote the graph-aware prompts for nodes $u$ and $v$, respectively. Taking Figure 2(c) as an example, assuming a pair of nodes $(u, v)$ connected by a $PP$ relation, representing a citation link, is sampled as the target edge. In this case, $PP$ relation is encoded as a relation token and placed between the graph-aware prompts of nodes $u$ and $v$, forming a relation-aware prompt that seamlessly integrates the citation interaction context.

## 4.3 HTRN-TAILORED PRETRAINING

Despite the generalizability of PLMs, fine-tuning is important for adapting their capabilities to unique characteristics of HTRNs. However, existing methods often overlook the design of HTRN-tailored pretraining tasks, where jointly considering textual information and structural heterogeneity is essential. As a result, they may limit their abilities to fully capture the complex interactions within HTRNs. To deal with this, we design two specialized *HTRN-tailored pretraining* tasks to enhance representation learning in HTRNs.

### 4.3.1 HGA-NSP

The HGA-NSP task extends the traditional NSP (Kenton & Toutanova, 2019) by capturing relation-specific semantics for the HTRN context. For a pair of nodes $(u, v)$ with a relation $r$, this task predicts whether $v$ connects $u$ via $r$. Positive samples are tuples $(u, r, v)$ where $u$ and $v$ are connected by $r$. For each positive sample, we generate the negative samples $(u, r, v')$ with a negative sampling ratio $nsr$, where $v'$ shares the same type as $v$, but not connected to $u$. The model calculates a probability $p$ for each sample to indicate the validity of the relation. The loss function is formally defined as:

$$\mathcal{L}_1 = -\frac{1}{N}\sum_{i=1}^{N}[y_i \log(p_i) + (1 - y_i)\log(1 - p_i)], \tag{1}$$

where $N$ is the total number of samples, $y_i$ is 1 for positive samples and 0 for negative samples.

### 4.3.2 HGA-MLM

The HGA-MLM task fine-tunes the model by predicting masked tokens in relation-aware prompts, covering both text and graph tokens. For each relation-aware prompt, a specified percentage of tokens are randomly masked based on the masking ratio $mr$, and the model predicts them using the remaining context. Different from the traditional MLM (Kenton & Toutanova, 2019) that relies solely on text, HGA-MLM incorporates both textual and heterogeneous structural information, which enables to jointly grasp both types of information, along with their interactions. The loss function is formally defined as:

$$\mathcal{L}_2 = -\frac{1}{M}\sum_{i=1}^{M} \log P(x_i|X_{\backslash i}), \tag{2}$$

where $M$ is the number of masked tokens, $x_i$ is the token at position $i$, $X_{\backslash i}$ denotes the input sequence with the $i$-th token masked. $P(x_i|X_{\backslash i})$ is the model's predicted probability for the correct token.

### 4.3.3 OVERALL LOSS

The total loss of HierPromptLM, $\mathcal{L}_{total} = \mathcal{L}_1 + \mathcal{L}_2$, combines HGA-NSP and HGA-MLM in a multi-task framework to capture node-level and edge-level heterogeneity. After fine-tuning the PLM with this loss, we use it to generate node/edge representations from graph/relation-aware prompts, facilitating downstream tasks such as node classification and link prediction. Pseudocode detailing offline processing, online training, and inference is in Appendix A.

**Time Complexity Analysis.** The time complexity of HierPromptLM is $O(TEL^2d)$, where $T, E, L, d$ are the number of Transformer layer, samples, input sequence length and hidden dimension, respectively. HierPromptLM has the same time complexity as existing fine-tuning LLM-based baselines (Zou et al., 2023; Jin et al., 2023b). Detailed analysis is provided in Appendix B.

## 5 EXPERIMENTS

### 5.1 EXPERIMENTAL SETTINGS

**Datasets.** We evaluate on three public real-world HTRN datasets (i.e. DBLP (Tang et al., 2008), OAG (Zhang et al., 2019b) and GoodReads (Wan & McAuley, 2018)), which are widely employed in prior works (Jin et al., 2023b; Zou et al., 2023). Details are in Appendix C.

**Baselines.** To evaluate the effectiveness of HierPromptLM, we compare it (and its training-free variant, HierPromptLM-frozen) against five groups of baselines: (1) homogeneous graph-based methods including GCN (Kipf & Welling, 2016) and GAT (Velickovic et al., 2017); (2) HG-based methods, comprising shallow models such as ComplEx (Trouillon et al., 2016), HIN2Vec (Fu et al., 2017), and M2V (Dong et al., 2017), as well as deep models involving RGCN (Schlichtkrull et al., 2018), RGAT (Busbridge et al., 2019), HGT (Hu et al., 2020), ie-HGCN (Yang et al., 2021b), and SHGP (Yang et al., 2022); (3) text-only PLMs including Bert (Kenton & Toutanova, 2019) and Albert (Chi et al., 2021); (4) HTRN-based methods including Heterformer (Jin et al., 2023b) and THLM (Zou et al., 2023); and (5) LLM-enhanced methods including TAPE (He et al., 2023). Groups (1) and (2) do not leverage text; following (Zou et al., 2023), we extend them with Bert-initialized node texts, yielding Method+Bert variants. Group (3) purely relies on text, Group (4) jointly models text and structures in HTRNs, and Group (5) integrates LLMs into graph learning. Reproducibility details are available in Appendix D.

### 5.2 PERFORMANCE COMPARISON

To evaluate the effectiveness, we conduct experiments on node classification and link prediction tasks, consistent with prior works (Zou et al., 2023; Jin et al., 2023b). Node classification is measured by Micro-F1 and Macro-F1, while link prediction is assessed with ROC-AUC, PR-AUC, and F1. To assess generalization, we also report a training-free variant, **HierPromptLM-frozen**, which feeds relation-aware prompts into a frozen PLM without fine-tuning. Results are reported in percentage (%), with further details in Appendix E. The results of graph classification task are in Appendix F.1.

**Node Classification.** As shown in Table 1, HierPromptLM consistently surpasses all baselines across all datasets, demonstrating significant performance improvement. This improvement stems from: (1) HierPromptLM jointly models text data and heterogeneous structures within a unified representation space, fully capturing their critical interactions within HTRNs. Notably, even without additional training, HierPromptLM-frozen, which relies solely on this prompt design framework, still outperforms all baselines, further validating its strength. (2) Two *HTRN-tailored pretraining tasks* facilitate deeper integration and understanding of both text and structures, thereby enhancing overall performance. Among the baselines, HTRN-based methods outperform PLM methods, confirming the benefit of combining text and structures. PLM methods generally outperform most graph-based models combined with Bert, as text plays a central role in node classification and naive fusion with structures often fails to preserve essential textual information, leading to limited performance.

**Link Prediction.** As observed in Table 2, HierPromptLM-frozen already almost outperforms all baselines. Built on this, HierPromptLM further improves performance by incorporating two *HTRN-tailored pretraining tasks*, consistently surpassing all approaches across datasets. Among the baselines, HTRN-based methods do not always outperform pure graph or text methods. This may

Table 1: Node classification. The best results are in bold; the second-best are underlined.

| Methods | DBLP | | OAG | | GoodReads | |
|---|---|---|---|---|---|---|
| | Micro-F1 | Macro-F1 | Micro-F1 | Macro-F1 | Micro-F1 | Macro-F1 |
| **Methods for Homogeneous Graphs** | | | | | | |
| GCN+Bert | $55.38_{\pm0.67}$ | $20.78_{\pm2.44}$ | $27.56_{\pm7.03}$ | $18.22_{\pm9.79}$ | $39.58_{\pm0.00}$ | $8.95_{\pm0.00}$ |
| GAT+Bert | $63.38_{\pm3.30}$ | $43.44_{\pm4.30}$ | $41.22_{\pm1.14}$ | $38.42_{\pm2.19}$ | $39.87_{\pm0.00}$ | $9.66_{\pm0.00}$ |
| **Shallow Methods for Heterogeneous Graphs** | | | | | | |
| ComplEx+Bert | $55.55_{\pm0.13}$ | $26.40_{\pm0.38}$ | $64.33_{\pm0.02}$ | $57.59_{\pm0.03}$ | $58.75_{\pm0.00}$ | $39.24_{\pm0.00}$ |
| HIN2Vec+Bert | $54.62_{\pm0.00}$ | $17.66_{\pm0.00}$ | $28.87_{\pm0.00}$ | $23.71_{\pm0.00}$ | $40.40_{\pm0.00}$ | $10.36_{\pm0.00}$ |
| M2V+Bert | $68.32_{\pm0.32}$ | $56.26_{\pm0.96}$ | $86.16_{\pm0.09}$ | $84.01_{\pm0.09}$ | $40.00_{\pm0.00}$ | $13.33_{\pm0.00}$ |
| **Deep Methods for Heterogeneous Graphs** | | | | | | |
| RGAT+Bert | $54.62_{\pm0.00}$ | $17.66_{\pm0.00}$ | $20.73_{\pm1.09}$ | $6.87_{\pm0.30}$ | $39.57_{\pm0.00}$ | $8.92_{\pm0.00}$ |
| RGCN+Bert | $86.73_{\pm0.00}$ | $83.61_{\pm0.00}$ | $23.06_{\pm0.00}$ | $11.09_{\pm0.00}$ | $39.58_{\pm0.00}$ | $8.96_{\pm0.00}$ |
| HGT+Bert | $87.90_{\pm0.00}$ | $85.28_{\pm0.00}$ | $84.84_{\pm1.37}$ | $84.59_{\pm1.37}$ | $69.82_{\pm0.00}$ | $49.34_{\pm0.00}$ |
| ie-HGCN+Bert | $53.36_{\pm2.51}$ | $18.29_{\pm1.33}$ | $29.74_{\pm6.02}$ | $27.08_{\pm5.59}$ | $42.30_{\pm0.00}$ | $12.74_{\pm0.00}$ |
| SHGP+Bert | $52.30_{\pm2.58}$ | $20.89_{\pm3.05}$ | $34.85_{\pm11.06}$ | $24.40_{\pm10.86}$ | $69.42_{\pm0.00}$ | $50.96_{\pm0.00}$ |
| **Text-only PLMs** | | | | | | |
| Bert | $81.67_{\pm1.53}$ | $77.30_{\pm2.29}$ | $85.99_{\pm1.52}$ | $85.78_{\pm1.56}$ | $71.82_{\pm0.00}$ | $54.43_{\pm0.00}$ |
| Albert | $79.08_{\pm1.57}$ | $73.82_{\pm2.87}$ | $85.01_{\pm1.07}$ | $84.86_{\pm1.17}$ | $70.17_{\pm0.00}$ | $52.31_{\pm0.00}$ |
| **Methods for Heterogeneous Text-rich Networks** | | | | | | |
| Heterformer | $86.40_{\pm0.08}$ | $83.10_{\pm0.12}$ | $93.82_{\pm0.28}$ | $90.61_{\pm0.29}$ | $72.35_{\pm0.00}$ | $60.40_{\pm0.00}$ |
| THLM | $85.21_{\pm1.07}$ | $82.12_{\pm1.46}$ | $94.83_{\pm0.56}$ | $92.18_{\pm0.87}$ | $73.30_{\pm0.00}$ | $59.67_{\pm0.00}$ |
| **LLM-enhanced Methods** | | | | | | |
| TAPE | $79.71_{\pm0.00}$ | $76.50_{\pm0.00}$ | $64.72_{\pm0.00}$ | $64.13_{\pm0.00}$ | $56.16_{\pm0.00}$ | $32.68_{\pm0.00}$ |
| **Our Methods** | | | | | | |
| HierPromptLM-frozen | $89.17_{\pm0.13}$ | $86.91_{\pm0.11}$ | $\mathbf{96.24_{\pm0.03}}$ | $94.17_{\pm0.06}$ | $76.27_{\pm0.00}$ | $64.10_{\pm0.00}$ |
| $\Delta_{frozen}$ | ↑3.21% | ↑4.58% | **↑1.48%** | ↑2.16% | ↑4.05% | ↑6.13% |
| HierPromptLM | $\mathbf{90.36_{\pm0.07}}$ | $\mathbf{88.15_{\pm0.09}}$ | $\mathbf{96.24_{\pm0.06}}$ | $\mathbf{94.43_{\pm0.09}}$ | $\mathbf{76.76_{\pm0.00}}$ | $\mathbf{64.72_{\pm0.00}}$ |
| $\Delta_{tuned}$ | **↑4.58%** | **↑6.08%** | **↑1.48%** | **↑2.44%** | **↑4.72%** | **↑7.15%** |

Table 2: Link prediction. The best results are in bold; the second-best are underlined.

| Methods | DBLP | | | OAG | | | GoodReads | | |
|---|---|---|---|---|---|---|---|---|---|
| | ROC-AUC | PR-AUC | F1 | ROC-AUC | PR-AUC | F1 | ROC-AUC | PR-AUC | F1 |
| **Methods for Homogeneous Graphs** | | | | | | | | | |
| GCN+Bert | $68.77_{\pm1.54}$ | $65.18_{\pm4.86}$ | $65.93_{\pm5.75}$ | $83.18_{\pm4.44}$ | $78.81_{\pm7.13}$ | $83.53_{\pm2.55}$ | $50.00_{\pm0.00}$ | $50.00_{\pm0.00}$ | $22.22_{\pm31.43}$ |
| GAT+Bert | $69.22_{\pm4.13}$ | $63.02_{\pm3.38}$ | $71.24_{\pm3.77}$ | $71.20_{\pm4.04}$ | $73.90_{\pm6.79}$ | $74.72_{\pm0.91}$ | $74.68_{\pm0.00}$ | $68.38_{\pm0.00}$ | $74.81_{\pm0.00}$ |
| **Shallow Methods for Heterogeneous Graphs** | | | | | | | | | |
| ComplEx+Bert | $66.38_{\pm0.03}$ | $61.46_{\pm0.06}$ | $63.03_{\pm0.16}$ | $77.86_{\pm0.02}$ | $72.77_{\pm0.07}$ | $76.44_{\pm0.04}$ | $75.75_{\pm0.01}$ | $69.51_{\pm0.01}$ | $75.74_{\pm0.01}$ |
| HIN2Vec+Bert | $50.55_{\pm0.00}$ | $50.28_{\pm0.00}$ | $50.37_{\pm0.00}$ | $66.96_{\pm0.00}$ | $64.44_{\pm0.00}$ | $55.44_{\pm0.00}$ | $70.17_{\pm0.00}$ | $63.71_{\pm0.00}$ | $71.92_{\pm0.00}$ |
| M2V+Bert | $78.36_{\pm0.16}$ | $72.57_{\pm0.48}$ | $77.93_{\pm0.56}$ | $84.04_{\pm0.27}$ | $80.11_{\pm0.87}$ | $83.09_{\pm0.18}$ | $72.36_{\pm2.57}$ | $66.79_{\pm1.65}$ | $70.38_{\pm5.28}$ |
| **Deep Methods for Heterogeneous Graphs** | | | | | | | | | |
| RGAT+Bert | $50.00_{\pm0.00}$ | $50.00_{\pm0.00}$ | $44.44_{\pm31.43}$ | $50.00_{\pm0.00}$ | $50.00_{\pm0.00}$ | $66.67_{\pm0.00}$ | $50.00_{\pm0.00}$ | $50.00_{\pm0.00}$ | $22.22_{\pm31.43}$ |
| RGCN+Bert | $71.86_{\pm0.50}$ | $65.85_{\pm0.63}$ | $71.51_{\pm0.04}$ | $53.20_{\pm0.02}$ | $51.66_{\pm0.01}$ | $66.89_{\pm0.37}$ | $50.46_{\pm0.00}$ | $50.23_{\pm0.00}$ | $66.65_{\pm0.00}$ |
| HGT+Bert | $71.20_{\pm5.01}$ | $66.24_{\pm5.40}$ | $70.99_{\pm5.80}$ | $84.19_{\pm0.45}$ | $81.62_{\pm1.39}$ | $82.51_{\pm1.50}$ | $74.22_{\pm5.20}$ | $69.86_{\pm2.50}$ | $69.32_{\pm12.59}$ |
| ie-HGCN+Bert | $71.09_{\pm3.83}$ | $66.51_{\pm4.13}$ | $68.16_{\pm7.68}$ | $68.27_{\pm3.88}$ | $63.00_{\pm3.87}$ | $67.45_{\pm3.77}$ | $82.64_{\pm0.54}$ | $79.32_{\pm3.18}$ | $81.20_{\pm3.20}$ |
| SHGP+Bert | $68.02_{\pm6.51}$ | $63.74_{\pm6.63}$ | $66.55_{\pm8.15}$ | $80.05_{\pm3.98}$ | $73.84_{\pm5.87}$ | $81.48_{\pm2.38}$ | $81.82_{\pm0.79}$ | $76.19_{\pm3.69}$ | $82.20_{\pm3.71}$ |
| **Text-only PLMs** | | | | | | | | | |
| Bert | $71.12_{\pm10.51}$ | $67.77_{\pm9.26}$ | $72.39_{\pm3.01}$ | $80.42_{\pm5.69}$ | $76.59_{\pm5.69}$ | $80.84_{\pm8.33}$ | $83.07_{\pm0.08}$ | $79.07_{\pm1.69}$ | $81.98_{\pm1.24}$ |
| Albert | $70.34_{\pm8.33}$ | $65.94_{\pm7.61}$ | $71.96_{\pm2.96}$ | $83.39_{\pm2.37}$ | $79.22_{\pm4.02}$ | $82.88_{\pm0.94}$ | $82.63_{\pm0.28}$ | $77.59_{\pm2.26}$ | $82.33_{\pm1.39}$ |
| **Methods for Heterogeneous Text-rich Networks** | | | | | | | | | |
| Heterformer | $72.04_{\pm0.91}$ | $70.27_{\pm1.09}$ | $75.46_{\pm0.77}$ | $79.45_{\pm0.25}$ | $76.49_{\pm0.52}$ | $80.91_{\pm0.35}$ | $74.83_{\pm0.00}$ | $68.58_{\pm0.00}$ | $74.82_{\pm0.00}$ |
| THLM | $72.60_{\pm3.36}$ | $68.18_{\pm4.18}$ | $70.13_{\pm4.06}$ | $79.53_{\pm1.28}$ | $73.35_{\pm1.64}$ | $79.76_{\pm1.26}$ | $80.96_{\pm0.09}$ | $75.45_{\pm1.12}$ | $80.65_{\pm1.18}$ |
| **LLM-enhanced Methods** | | | | | | | | | |
| TAPE | $78.21_{\pm0.00}$ | $73.92_{\pm0.00}$ | $75.94_{\pm0.00}$ | $82.75_{\pm0.00}$ | $82.30_{\pm0.00}$ | $79.39_{\pm0.00}$ | $81.10_{\pm0.00}$ | $76.98_{\pm0.00}$ | $79.54_{\pm0.00}$ |
| **Our Methods** | | | | | | | | | |
| HierPromptLM-frozen | $81.04_{\pm0.00}$ | $75.78_{\pm0.01}$ | $80.43_{\pm0.02}$ | $85.42_{\pm0.01}$ | $80.56_{\pm0.06}$ | $85.25_{\pm0.06}$ | $84.05_{\pm0.01}$ | $78.64_{\pm0.00}$ | $84.04_{\pm0.01}$ |
| $\Delta_{frozen}$ | ↑3.42% | ↑2.52% | ↑3.21% | ↑1.47% | ↓2.16% | ↑2.06% | ↑1.18% | ↓0.86% | ↑2.08% |
| HierPromptLM | $\mathbf{85.56_{\pm0.04}}$ | $\mathbf{80.44_{\pm0.01}}$ | $\mathbf{85.56_{\pm0.05}}$ | $\mathbf{89.24_{\pm0.01}}$ | $\mathbf{84.92_{\pm0.07}}$ | $\mathbf{89.27_{\pm0.01}}$ | $\mathbf{86.91_{\pm0.01}}$ | $\mathbf{81.94_{\pm0.01}}$ | $\mathbf{86.98_{\pm0.00}}$ |
| $\Delta_{tuned}$ | **↑9.19%** | **↑8.82%** | **↑9.79%** | **↑6.00%** | **↑3.18%** | **↑6.87%** | **↑4.62%** | **↑3.30%** | **↑5.65%** |

be attributed to the fact that existing HTRN approaches treat text and structure separately, failing to capture their critical interactions. This provides further evidence for the importance of joint modeling and highlights the superiority of our proposed HierPromptLM.

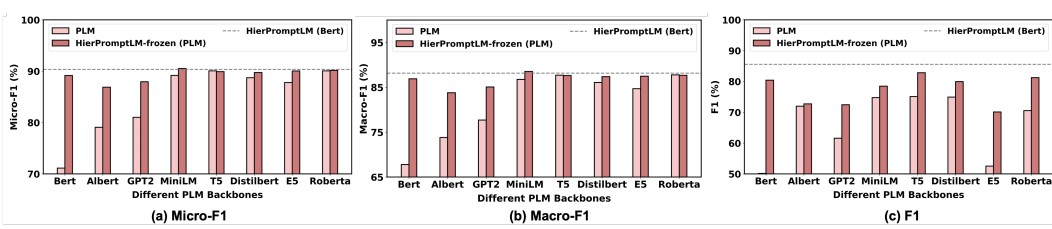

Figure 3: Different PLM backbones on the DBLP dataset.

## 5.3 EXTENSION TO DIFFERENT PLM BACKBONES

To evaluate the generalization ability of HierPromptLM, we replace its PLM backbone with various PLMs: Bert (Kenton & Toutanova, 2019), Albert (Chi et al., 2021), GPT2 (Radford et al., 2019), MiniLM (Wang et al., 2020), T5 (Raffel et al., 2020), Distilbert (Sanh, 2019), Roberta (Liu, 2019b), and E5 (Wang et al., 2023). We conduct this evaluation on the DBLP dataset under the training-free setting, resulting in a set of **HierPromptLM-frozen (PLM)** variants. The fine-tuned **HierPromptLM (Bert)** is included as a reference. Figure 3 presents Micro-F1 and Macro-F1 for node classification, and F1 for link prediction; ROC-AUC and PR-AUC results are provided in Appendix F.2. Across all backbones, HierPromptLM-frozen (PLM) generally outperforms the corresponding PLMs, demonstrating the advantages of our hierarchical prompt for HTRNs. These results highlight the adaptability of HierPromptLM to different PLM backbones, and even with a small PLM such as Bert, **HierPromptLM (Bert)** consistently surpasses much larger PLMs. Additional results with different backbones under the training setting are reported in Appendix F.3, showing similar trends.

## 5.4 ABLATION STUDY

To assess the impact of key components in HierPromptLM, we perform ablation study on DBLP, focusing on: (1) **w/o HGA-MLM** removes the HGA-MLM task; (2) **w/o HGA-NSP** eliminates the HGA-NSP task; (3) **w/o GraphToken** deletes meta-path-based graph tokens from graph-aware prompts, using only text data; (4) **w/o NodeSpecific** excludes node-specific information (*i.e.,* abstracts) in graph-aware prompts. As shown in Table 3, we observe: (1) Removing HGA-MLM substantially degrades both node classification and link prediction, showing its necessity for capturing textual and structural information in HTRNs. (2) HierPromptLM consistently outperforms w/o HGA-NSP, confirming the benefits of this pretraining task in modeling relation heterogeneity. (3) HierPromptLM's advantage over w/o GraphToken confirms the strength of meta-path-based subgraphs in enhancing node representations by augmenting structural information. (4) w/o Node-Specific underperforms on node classification but surpasses HierPromptLM on link prediction, indicating that abstract information aids in node discrimination but may introduce noise for link prediction, making its utility task-dependent. Appendix F.6 further validates HierPromptLM's robustness with different subgraph generation strategies, considering large neighborhoods and absent meta-paths scenarios.

Table 3: Ablation study on the DBLP dataset.

| Methods | Node classification | | Link Prediction | | |
| --- | --- | --- | --- | --- | --- |
| | Micro-F1 | Macro-F1 | ROC-AUC | PR-AUC | F1 |
| w/o HGA-MLM | $89.10_{\pm 0.10}$ | $86.82_{\pm 0.18}$ | $82.73_{\pm 0.00}$ | $77.36_{\pm 0.00}$ | $82.51_{\pm 0.00}$ |
| w/o HGA-NSP | $89.12_{\pm 0.10}$ | $86.78_{\pm 0.21}$ | $81.13_{\pm 0.00}$ | $75.78_{\pm 0.02}$ | $80.64_{\pm 0.02}$ |
| w/o GraphToken | $87.45_{\pm 0.11}$ | $84.76_{\pm 0.15}$ | $80.15_{\pm 0.07}$ | $75.56_{\pm 0.02}$ | $78.74_{\pm 0.15}$ |
| w/o NodeSpecific | $86.22_{\pm 0.12}$ | $82.84_{\pm 0.23}$ | $\mathbf{86.25_{\pm 0.25}}$ | $\mathbf{81.55_{\pm 0.14}}$ | $\mathbf{86.10_{\pm 0.35}}$ |
| HierPromptLM | $\mathbf{90.36_{\pm 0.07}}$ | $\mathbf{88.15_{\pm 0.09}}$ | $85.56_{\pm 0.04}$ | $80.44_{\pm 0.01}$ | $85.56_{\pm 0.05}$ |

## 5.5 FURTHER ANALYSIS OF EFFICIENCY, DOMAIN SHIFT AND PARAMETERS

HierPromptLM achieves comparable FPS and memory usage to existing HTRN methods (Appendix F.4), surpasses all baselines when transferred from DBLP to OAG and GoodReads (Appendix F.5). We further analyze the impact of three key parameters in Appendix F.7.

## 6 CONCLUSION

In this paper, we introduce HierPromptLM, a novel PLM-based method for HTRN representation learning, which integrates text and heterogeneous structures within a unified space. It has two key components: (1) an automated textualization mechanism that employs prompt learning to integrate both types of data at node and edge levels, and (2) two *HTRN-tailored pretraining* tasks for fine-tuning. Comprehensive experiments demonstrate that HierPromptLM outperforms state-of-the-art methods.

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

---

**Algorithm 1** HierPromptLM: Offline Processing and Online Training

---

1: **Input:** An HTRN: $\mathcal{G} = (\mathcal{V}, \mathcal{E}, \mathcal{C}, \mathcal{R}, \mathcal{T})$, a meta-path set: $\mathcal{M}$, a meta-path-based description set: $\mathcal{MD}$, a frozen PLM: $\Phi_1$, a tunable PLM: $\Phi_2$
2: **Output:** The fine-tuned PLM: $\Phi_2'$

    **Step 1: Offline Processing - Precomputing Graph-Aware Prompts**
3: Initialize an empty set $\mathcal{F}$ to store graph-aware prompts for all nodes
4: **for** each node $u \in \mathcal{V}$ **do**
5:    Initialize an empty set $\mathcal{S}(u)$ to store subgraph summaries for $u$
6:    **for** each meta-path $m \in \mathcal{M}$ **do**
7:      Extract the $m$-based subgraph $\mathcal{G}_m(u)$ for $u$
8:      Generate a textual summary $\mathcal{S}_m(u)$ for $\mathcal{G}_m(u)$ via a given subgraph program function
9:      Add $\mathcal{S}_m(u)$ to $\mathcal{S}(u)$
10:    **end for**
11:    Distill graph summaries in $\mathcal{S}(u)$ into corresponding graph tokens $\mathcal{T}(u)$ via $\Phi_1$
12:    Generate $u$'s graph-aware prompt $\mathcal{F}(u)$ by combining its text data, $\mathcal{T}(u)$ and $\mathcal{MD}$
13:    Add $\mathcal{F}(u)$ to $\mathcal{F}$
14: **end for**
15: **Return:** The graph-aware prompt set $\mathcal{F}$

    **Step 2: Online Training - Relation-Aware Prompt Construction and Fine-Tuning**
16: Initialize an empty set $\mathcal{H}$ to store relation-aware prompts for all training edges
17: Initialize learnable relation tokens $\mathcal{RT}$ for all edge types
18: Add $\mathcal{RT}$ to $\Phi_2$'s vocabulary
19: **for** each edge $e = (u, r, v) \in \mathcal{E}_{train}$ **do**
20:    Retrieve graph-aware prompts $\mathcal{F}(u)$ and $\mathcal{F}(v)$ from $\mathcal{F}$ for $u$ and $v$
21:    Retrieve the learnable relation token $\mathcal{RT}(r)$ from $\mathcal{RT}$ for $r$
22:    Generate $e$'s relation-aware prompt $\mathcal{H}(e)$ by concatenating $\mathcal{F}(u)$, $\mathcal{RT}(r)$ and $\mathcal{F}(v)$
23:    Add $\mathcal{H}(e)$ to $\mathcal{H}$
24: **end for**
25: Fine-tune $\Phi_2$ with $\mathcal{H}$ based on the loss $\mathcal{L}_{total}$
26: **Return:** The fine-tuned PLM $\Phi_2'$

---

# A   PSEUDOCODE FOR HIERPROMPTLM

In our HierPromptLM, online training involves fine-tuning a tunable PLM with relation-aware prompts based on the overall loss $\mathcal{L}_{total}$, while the generation for graph-aware prompt can be precomputed offline for efficiency enhancement. The pseudocodes of these two processes are provided in Algorithm 1. Moreover, Algorithm 2 details the inference process, which generates both node and edge embeddings using the fine-tuned HierPromptLM. Subsequently, these learned embeddings can be utilized for downstream tasks such as node classification and link prediction.

## A.1   OFFLINE PROCESSING

The offline processing phase precomputes graph-aware prompts to improve HierPromptLM's overall efficiency. For each node $u$, meta-path-based subgraphs are extracted using predefined meta-paths, capturing the essential structures necessary to represent heterogeneous relations in HTRNs. Textual summaries are then generated for each subgraph using a pre-defined subgraph program function $\mathcal{P}(\cdot)$ as discussed in Section 4.1.1 and distilled into compact graph tokens via a frozen PLM $\Phi_1$. These graph tokens with their corresponding descriptions, combined with the node's own text data, constitute $u$'s graph-aware prompt $\mathcal{F}(u)$. Precomputing these prompts significantly reduces the computational burden during online training, enabling the framework to scale effectively for large and complex HTRNs.

---

**Algorithm 2** Inference (Testing Phase)

---

1: **Input:** A fine-tuned PLM $\Phi_2'$, graph-aware prompts $\mathcal{F}$, testing nodes $\mathcal{V}_{test}$, testing edges $\mathcal{E}_{test}$
2: **Output:** Node embeddings $Z_V$ and edge embeddings $Z_E$

    **Node Embedding Generation**
3: **for** each node $u \in \mathcal{V}_{test}$ **do**
4:     Retrieve the graph-aware prompt $\mathcal{F}(u)$ from $\mathcal{F}$
5:     Obtain the node embedding $z_u$ by feeding $\mathcal{F}(u)$ into $\Phi_2'$
6:     Add $z_u$ to $Z_V$
7: **end for**
8: **Return:** Node embeddings $Z_V$

    **Edge Embedding Generation**
9: **for** each edge $e = (u, r, v) \in \mathcal{E}_{test}$ **do**
10:     Retrieve graph-aware prompts $\mathcal{F}(u)$ and $\mathcal{F}(v)$ from $\mathcal{F}$ for nodes $u$ and $v$
11:     Retrieve the learnable relation token $\mathcal{RT}(r)$ from $\Phi_2$'s vocabulary for $r$
12:     Construct the relation-aware prompt $\mathcal{H}(e)$ by concatenating $\mathcal{F}(u)$, $\mathcal{RT}(r)$, and $\mathcal{F}(v)$
13:     Obtain the edge embedding $z_e$ by feeding $\mathcal{H}(e)$ into $\Phi_2'$
14:     Add $z_e$ to $Z_E$
15: **end for**
16: **Return:** Edge embeddings $Z_E$

---

## A.2 ONLINE TRAINING

The online training phase leverages precomputed graph-aware prompts to construct relation-aware prompts, which are used to fine-tune the tunable PLM $\Phi_2$. For each edge $e = (u, r, v)$ in the training set $\mathcal{E}_{train}$, the graph-aware prompts of nodes $u$ and $v$ are retrieved and combined with a learnable relation token representing the edge type $r$, forming the relation-aware prompt for $e$. These relation-aware prompts are then fed into $\Phi_2$, which is fine-tuned with the loss $\mathcal{L}_{total}$ as discussed in Section 4.3.3. This process effectively captures the heterogeneity and text-rich characteristics of HTRNs, facilitating various downstream applications.

## A.3 INFERENCE

Once fine-tuned, HierPromptLM is employed to generate node embeddings from graph-aware prompts and edge embeddings from relation-aware prompts for testing nodes and edges. These embeddings can be directly utilized for downstream tasks, such as node classification and link prediction.

**Node Embedding Generation.** In the inference phase, node embeddings are generated using the precomputed graph-aware prompts $\mathcal{F}$. For each testing node $u \in \mathcal{V}_{test}$, the corresponding graph-aware prompt $\mathcal{F}(u)$ is first retrieved from $\mathcal{F}$. This prompt is then fed into the fine-tuned PLM $\Phi_2'$, producing the node embedding $z_u$ for $u$, which is subsequently added to the node embedding set $Z_V$. After processing all testing nodes, the complete node embedding set $Z_V$ is returned for tasks such as node classification.

**Edge Embedding Generation.** Similarly, we leverage the fine-tuned PLM $\Phi_2'$ to generate edge embeddings from relation-aware prompts. For each testing edge $e = (u, r, v) \in \mathcal{E}_{test}$, the graph-aware prompts $\mathcal{F}(u)$ and $\mathcal{F}(v)$ for nodes $u$ and $v$ are retrieved from the precomputed graph-aware prompt set $\mathcal{F}$. These prompts are then combined with the learnable relation token $\mathcal{RT}(r)$, corresponding to the edge $r$, which is retrieved from the vocabulary of $\Phi_2'$. Finally, these components are concatenated to form the relation-aware prompt $\mathcal{H}(e)$, which is fed into $\Phi_2'$ to generate the edge embedding $z_e$. Subsequently, this embedding $z_e$ is added to the edge embedding setting $Z_E$. After processing all testing edges, the complete set $Z_E$ is returned for downstream tasks such as link prediction.

Table 4: Dataset statistics.

| Dataset | Objects (#) | #Object | Relations | #Relation | #Label Type | #Labeled Object |
|---|---|---|---|---|---|---|
| DBLP | $P$(53,614), $A$(10,279), $V$(3,524) | 67,417 | $P \rightleftharpoons P, P \rightleftharpoons A, P \rightleftharpoons V$ | 149,545 | 4 | 53,614 |
| OAG | $P$(41,632), $A$(20,366), $F$(20), $I$(1,790) | 63,808 | $P \rightleftharpoons P, P \rightleftharpoons A, P \rightleftharpoons F, A \rightleftharpoons I$ | 240,219 | 5 | 41,632 |
| GoodReads | $B$(120,207), $A$(69,323), $P$(19,004) | 208,534 | $B \rightleftharpoons B, B \rightleftharpoons A, B \rightleftharpoons P$ | 511,470 | 10 | 120,207 |

## B   TIME COMPLEXITY ANALYSIS

The time complexity of HierPromptLM is $O(TEL^2d)$, where $T$, $E$, $L$, and $d$ denote the number of Transformer layer, the number of samples (both positive and negative), the input sequence length, and the feature dimension, respectively.

In our HierPromptLM, relation-aware prompts are constructed by combining the graph-aware prompts of two nodes (precomputed offline) with a relation token, which requires only simple concatenation with a negligible complexity of $O(L)$. The primary computational cost arises from the Transformer Self-Attention mechanism in the PLM, which has a complexity of $O(TL^2d)$ per sample. Thus, the overall complexity is $O(TEL^2d)$, scaling linerly with $E$ and quadratically with $L$.

This scalability is further optimized by our flexible training design: not all edges in the HTRN need to be included for training, as the number of positive samples and the negative sampling ratio are adjustable. In addition, the utilization of graph tokens significantly reduces $L$ by distilling heterogeneous graph structures into compact graph tokens offline.

## C   DATASET DESCRIPTION

We conduct experiments on three public real-world HTRN datasets (i.e., DBLP[1], OAG[2] and GoodReads[3]), which are widely employed in prior works (Jin et al., 2023b; Zou et al., 2023). The main statistics of datasets used are summarized in Table 4 and the details are described as follows.

- **DBLP (Tang et al., 2008).** The dataset is extracted from four fields of DBLP bibliography. The four fields are: mathematical optimization, pattern recognition, computer vision and computer network. Based on the dataset, we construct an HTRN containing three types of nodes: 53,614 papers (P), 10,279 authors (A) and 3,524 venues (V), and three types of relations: 50,330 paper-paper (PP) edges, 47,573 paper-author (PA) edges and 51,642 paper-venue (PV) edges. We treat papers as text-rich nodes and extract the title and abstract parts as their textual information. Authors and venues are regarded as textless nodes with only name information. The target nodes, papers, are labeled according to the four fields.

- **OAG (Zhang et al., 2019b).** The dataset is a subgraph extracted from OAG based on five venues: IEEE Transactions on Biomedical Engineering, IEEE Transactions on Vehicular Technology, IEEE Journal of Solid State Circuits, INTERACT and SIGGRAPH. Based on the dataset, we construct an HTRN containing four types of nodes: 41,632 papers (P), 20,366 authors (A) and 20 fields (F), 1,790 institutions (I) and four types of relations: 55,794 paper-paper (PP) edges, 73,584 paper-author (PA) edges 87,972 paper-field (PF) edges and 22,869 author-institution (AI) edges. Papers are treated as text-rich nodes, with titles and abstracts serving as their textual information. Authors, fields and institutions are considered textless nodes with only name information. The target nodes, papers, are labeled based on the five venues.

- **GoodReads (Wan & McAuley, 2018).** We construct a subgraph from the GoodReads dataset following (Wan & McAuley, 2018; Zou et al., 2023), containing three types of nodes: books (B), authors (A), and publishers (P)—and three relation types: 248,735 book–book (BB) edges, 170,438 book–author (BA) edges, and 92,297 book–publisher (BP) edges. Books are treated as rich-text nodes, with their brief introductions used as textual descriptions. Authors and publishers are considered textless nodes, and their names are used as text attributes. The dataset includes ten

---

[1]`https://originalstatic.aminer.cn/misc/dblp.v12.7z`

[2]`https://github.com/UCLA-DM/pyHGT/`

[3]`https://sites.google.com/eng.ucsd.edu/ucsdbookgraph/home`

genres: "children", "comics, graphic", "fantasy, paranormal", "fiction", "history, historical fiction, biography", "mystery, thriller, crime", "non-fiction", "poetry", "romance", "young-adult". Each book is categorized into one or more genres, forming a multi-label classification task.

## D    REPRODUCIBILITY

To ensure a fair comparison with prior work (Zou et al., 2023; Jin et al., 2023b), we adopt Bert (110M parameters (Kenton & Toutanova, 2019)) as the default backbone PLM. We initialize the model with weights in Bert checkpoint released from Transformers tools[4]. HierPromptLM is optimized using AdamW (Loshchilov, 2017) with a learning rate searched in [1e-5, 5e-5]. We set the default masking ratio to 0.15 and the negative sampling ratio to 1. For consistency, all baselines use a hidden dimension of 768 and follow the recommended hyperparameters from their original papers. For the LLM-enhanced baseline TAPE (He et al., 2023), we use Llama-2[5] (7B, (Chi et al., 2021)) as LLM augmentation and Bert as the PLM backbone. All models are trained for 10 times and, the mean and standard variance of test performance are reported.

For the extensions to different PLM-based backbones in the training-free setting, we follow the official configuration of the following models to ensure a fair comparison: Albert[6] (12M, (Chi et al., 2021)), GPT2[7] (124M, (Radford et al., 2019)), MiniLM[8] (22M, (Wang et al., 2020)), T5[9] (220M, (Raffel et al., 2020)), Distilbert[10] (66M, (Sanh, 2019)), E5[11] (335M, (Wang et al., 2023)) and Roberta[12] (355M, (Liu, 2019b)). To ensure reproducibility, we include our source code, datasets, as well as the instructions to the selected baselines, in an anonymous repository[13]. All experiments are performed using two NVIDIA RTX A5000 GPUs, each with 24GB of memory.

## E    SETTINGS

### E.1    NODE CLASSIFICATION

Node classification aims to assign categories to nodes within a network. Following (Hu et al., 2020), we train a separate linear Support Vector Machine (LinearSVM) (Fan et al., 2008) using the node embeddings generated by our model as input, with the corresponding node labels serving as the classification targets. For DBLP and OAG, we use 70% of the nodes for training, 10% for validation, and 20% for testing. For GoodReads, the split is 20%, 10%, and 70%, respectively. Following (Zou et al., 2023; Jin et al., 2023b), we evaluate performance using Micro-F1 and Macro-F1 scores, reported as percentages (%).

### E.2    LINK PREDICTION

Link prediction aims to predict missing edges in a network. In HTRNs, there are various heterogeneous edges, and missing links can belong to any of these types. Therefore, to comprehensively evaluate the model performance, multiple relation types (e.g., paper-paper, paper-author, paper-venue for the DBLP dataset; paper-paper, paper-author, paper-field, author-institution for the OAG dataset; book-book, book-author, book-publisher for the GoodReads dataset) are considered simultaneously, allowing a thorough assessment of the model's ability to capture diverse interactions in HTRNs, as discussed in Section 5.2. Following (Zou et al., 2023), we adopt a standard sampling strategy for link prediction, using 10% of edges for training and 40% for testing on the DBLP and OAG datasets. For the GoodReads dataset, we randomly sample 2,000 training edges and 8,000 testing edges for each

---

[4]https://huggingface.co/google-bert
[5]https://huggingface.co/meta-llama/Llama-2-7b-chat-hf
[6]https://huggingface.co/albert/albert-base-v2
[7]https://huggingface.co/openai-community/gpt2
[8]https://huggingface.co/sentence-transformers/all-MiniLM-L6-v2
[9]https://huggingface.co/google-t5/t5-base
[10]https://huggingface.co/sentence-transformers/multi-qa-distilbert-cos-v1
[11]https://huggingface.co/embaas/sentence-transformers-e5-large-v2
[12]https://huggingface.co/sentence-transformers/all-roberta-large-v1
[13]https://anonymous.4open.science/r/HierPromptLM-code

type of relation. Evaluation metrics, including ROC-AUC, PR-AUC, and F1 scores, are adopted in line with prior work (Liu et al., 2022).

Table 5: Graph classification on DBLP. The best results are in bold; the second-best are underlined.

| Methods | Micro-F1 | Macro-F1 |
|---|---|---|
| **Methods for Homogeneous Graphs** | | |
| GCN+Bert | $84.93_{\pm 6.78}$ | $81.000_{\pm 9.54}$ |
| GAT+Bert | $65.43_{\pm 2.01}$ | $65.20_{\pm 3.20}$ |
| **Shallow Methods for Heterogeneous Graphs** | | |
| ComplEx+Bert | $47.00_{\pm 0.00}$ | $48.11_{\pm 0.00}$ |
| HIN2Vec+Bert | $20.90_{\pm 0.00}$ | $20.52_{\pm 0.00}$ |
| M2V+Bert | $76.10_{\pm 0.00}$ | $76.36_{\pm 0.00}$ |
| **Deep Methods for Heterogeneous Graphs** | | |
| RGAT+Bert | $20.00_{\pm 0.00}$ | $6.67_{\pm 0.00}$ |
| RGCN+Bert | $40.43_{\pm 0.42}$ | $36.76_{\pm 0.53}$ |
| HGT+Bert | $85.47_{\pm 0.49}$ | $85.24_{\pm 0.47}$ |
| ie-HGCN+Bert | $90.60_{\pm 3.89}$ | $90.27_{\pm 4.23}$ |
| SHGP+Bert | $91.20_{\pm 3.00}$ | $91.20_{\pm 3.02}$ |
| **Text-only PLMs** | | |
| Bert | $76.57_{\pm 1.93}$ | $75.92_{\pm 2.71}$ |
| Albert | $86.00_{\pm 0.70}$ | $85.85_{\pm 0.42}$ |
| **Methods for Heterogeneous Text-rich Networks** | | |
| Heterformer | $54.58_{\pm 0.12}$ | $42.71_{\pm 0.38}$ |
| THLM | $67.37_{\pm 1.72}$ | $67.68_{\pm 1.61}$ |
| **LLM-enhanced Methods** | | |
| TAPE | $87.90_{\pm 0.00}$ | $87.84_{\pm 0.00}$ |
| **Our Methods** | | |
| HierPromptLM-frozen | $\mathbf{93.83_{\pm 0.05}}$ | $\mathbf{93.82_{\pm 0.05}}$ |
| $\Delta_{frozen}$ | ↑ **2.88%** | ↑ **2.87%** |
| HierPromptLM | $\underline{93.70_{\pm 0.00}}$ | $\underline{93.62_{\pm 0.05}}$ |
| $\Delta_{tuned}$ | ↑ 2.74% | ↑ 2.65% |

# F  ADDITIONAL EXPERIMENTS

## F.1  GRAPH CLASSIFICATION

In addition to node-level and edge-level tasks (i.e., node classification and link prediction), we further evaluate a graph-level task on the DBLP dataset, namely meta-path-based subgraph classification. Graph classification aims to categorize subgraphs according to their structural patterns. Specifically, we consider meta-paths PA, PP, PV, VP, and AP. For each meta-path, subgraphs are constructed by identifying the target node and its neighbors connected via the meta-path, and the subgraph is labeled by the corresponding meta-path type (i.e., PA, PP, PV, VP, and AP). We randomly sample 200 subgraphs per meta-path for training and testing, resulting in 1,000 training and 1,000 testing samples. After generating node embeddings with our model, we construct subgraph embeddings by applying mean pooling over all nodes contained within each subgraph. These subgraph embeddings are then used as input features to a LinearSVM for prediction. Performance is evaluated using Micro-F1 and Macro-F1 scores, reported as percentages (%). The results are shown in Table 5.

As shown in Table 5, we make several key observations: (1) Both HierPromptLM and HierPromptLM-frozen outperform all baselines, underscoring the effectiveness of our framework. (2) HierPromptLM-frozen achieves the overall best performance, even surpassing the fine-tuned version, indicating that our prompt-based textualization framework alone is sufficient to unify text and structure into highly discriminative representations without additional training. (3) In contrast, shallow heterogeneous methods such as ComplEx, HIN2Vec, and M2V perform poorly, reflecting their limited ability to capture the complex semantic and structural dependencies required for graph-level classification. (4)

HTRN-based methods underperform even compared to text-only PLMs, as they process text and structure separately and therefore fail to capture their critical interactions in a unified space.

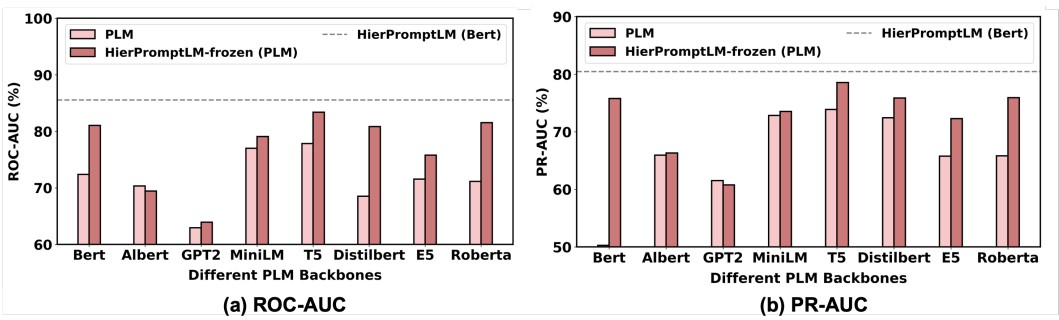

Figure 4: Different PLM backbones on the DBLP dataset.

## F.2 EXTENSION TO DIFFERENT PLM BACKBONES

Figure 5 reports ROC-AUC and PR-AUC results for link prediction on the DBLP dataset. Across all backbones, we observe that HierPromptLM-frozen (PLM) consistently outperforms its standalone PLM counterparts, highlighting the effectiveness of our relation-aware prompt for HTRN modeling. These results underscore the robustness and adaptability of our HierPromptLM. More importantly, the fine-tuned **HierPromptLM (Bert)** consistently and significantly outperforms all methods across both metrics, further validating the impact of our proposed *HTRN-tailored pretraining* tasks.

## F.3 EVALUATING PLM BACKBONES FOR GRAPH-TOKEN DISTILLATION

To further assess the generalization of HierPromptLM in the training setting, we replace the frozen PLM backbone in our HierPromptLM with alternatives such as T5 and GPT2 during the offline graph-token distillation stage, resulting in HierPromptLM (T5) and HierPromptLM (GPT2) variants. Importantly, the subsequent training configuration is kept identical across all settings to ensure fair comparison. We focus on replacing frozen PLM backbones for offline processing, as fine-tuning large models is time-intensive and can be deferred for future work. The results for both tasks on the DBLP dataset are presented in Figure 5. As shown, we can see that HierPromptLM (T5) and HierPromptLM (GPT2) outperform the Bert-based variant, attributed to the larger capacity and stronger representation power of T5 and GPT2. These findings validate the benefit of leveraging more powerful PLMs and highlight HierPromptLM's adaptability to diverse backbones while maintaining strong performance.

## F.4 EFFICIENCY EVALUATION

To assess training efficiency, we primarily use FPS (frames per second), which measures how many samples are processed per second during training. A higher FPS indicates more efficient training. As shown in Table 6, HierPromptLM achieves the highest FPS on DBLP (24.42) and OAG (24.32), while

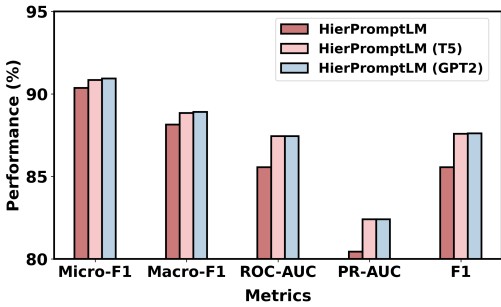

Figure 5: Different PLM backbones for graph-token distillation on the DBLP dataset.

Table 6: Efficiency evaluation.

| Methods | DBLP | | | | OAG | | | | GoodReads | | | |
|---|---|---|---|---|---|---|---|---|---|---|---|---|
| | ↓ Memory | #Sample | ↓ Time | ↑ FPS | ↓ Memory | #Sample | ↓ Time | ↑ FPS | ↓ Memory | #Sample | ↓ Time | ↑ FPS |
| Heterformer | 1405 | 23802 | 5777 | 4.12 | 1468 | 16708 | 1237 | 13.51 | 3158 | 8028 | 265 | 30.25 |
| THLM | 2257 | 53614 | 2440 | 21.97 | 2042 | 41632 | 2448 | 17.01 | 2453 | 120207 | 5085 | 23.64 |
| HierPromptLM | 1624 | 257430 | 10542 | 24.42 | 1867 | 275784 | 11344 | 24.32 | 1383 | 829596 | 34641 | 23.95 |

maintaining competitive performance on GoodReads (23.95), all with memory usage comparable to two baselines specifically designed for HTRN representation learning.

Table 7: Domain shift evaluation from the DBLP dataset to the OAG dataset.

| Methods | Node classification | | Link Prediction | | |
|---|---|---|---|---|---|
| | Micro-F1 | Macro-F1 | ROC-AUC | PR-AUC | F1 |
| HierPromptLM(DBLP) | $96.02_{\pm0.03}$ | $93.90_{\pm0.06}$ | $87.99_{\pm0.03}$ | $83.55_{\pm0.17}$ | $87.95_{\pm0.03}$ |
| Best Baselines | $94.83_{\pm0.56}$ | $92.18_{\pm0.87}$ | $84.19_{\pm0.45}$ | $82.30_{\pm0.00}$ | $83.53_{\pm2.55}$ |
| HierPromptLM | $96.24_{\pm0.06}$ | $94.43_{\pm0.09}$ | $89.24_{\pm0.01}$ | $84.92_{\pm0.07}$ | $89.27_{\pm0.01}$ |

Table 8: Domain shift evaluation from the DBLP dataset to the GoodReads dataset.

| Methods | Node classification | | Link Prediction | | |
|---|---|---|---|---|---|
| | Micro-F1 | Macro-F1 | ROC-AUC | PR-AUC | F1 |
| HierPromptLM(DBLP) | $74.55_{\pm0.00}$ | $61.06_{\pm0.00}$ | $84.85_{\pm0.00}$ | $79.33_{\pm0.00}$ | $85.00_{\pm0.00}$ |
| Best Baselines | $73.30_{\pm0.00}$ | $59.67_{\pm0.00}$ | $83.07_{\pm0.08}$ | $79.32_{\pm3.18}$ | $82.33_{\pm1.99}$ |
| HierPromptLM | $76.76_{\pm0.00}$ | $64.72_{\pm0.00}$ | $86.91_{\pm0.01}$ | $81.94_{\pm0.01}$ | $86.98_{\pm0.00}$ |

## F.5   DOMAIN SHIFT ANALYSIS

To further evaluate cross-dataset transferability, we conduct zero-shot transfer experiments where our HierPromptLM pretrained on the DBLP dataset is directly tested on other datasets—specifically OAG and GoodReads—without any additional training or fine-tuning. We denote this setting as **HierPromptLM(DBLP)**. For comparison, we report the best-performing baselines pretrained or trained on the target datasets, referred to as **Best Baselines**. The results on the OAG and GoodReads datasets are shown in Table 7 and Table 8, respectively.

As observed, we have two main observations: (1) HierPromptLM(DBLP) consistently outperforms the best baselines on both datasets, despite not being trained or fine-tuned on them. This demonstrates strong transferability and confirms that our model generalizes well under domain shift. (2) On the GoodReads dataset, which represents a non-citation, non-academic HTRN with different node types, relations, and label semantics, HierPromptLM(DBLP) still achieves substantial gains. This highlights the broad generalization ability of our framework across different domains.

## F.6   SUBGRAPH GENERATION STRATEGIES

**Meta-path Sampling.** When the number of candidate meta-paths is large, using all of them to build subgraphs is both computationally costly and potentially redundant. Instead, we sample from a candidate pool of meta-paths rather than exhaustively including all possibilities, which reduces complexity while preserving diverse structural patterns. Specifically, for the DBLP dataset, we consider a candidate pool {PP, PA, PV, AP, VP, PAP, PVP, APA, PAPAP, APAPA, VPVPV}. The results for different meta-path combinations under the training-free setting are reported in Table 9. As observed, our model consistently outperforms the best baseline across different meta-path combinations, demonstrating that the model effectively benefits from meta-path sampling while maintaining strong and robust performance.

**Neighbor Sampling.** When a meta-path connects to a large number of neighbors, using all of them also can be computationally expensive. To address this, we randomly sample a subset of neighbors for each meta-path and construct the corresponding meta-path-based subgraph summary from the sampled set. This strategy effectively controls neighborhood size and reduces computational

Table 9: Results of meta-path and k-hop sampling on the DBLP dataset.

| Meta-path | Node classification | | Link Prediction | | |
| --- | --- | --- | --- | --- | --- |
| | Micro-F1 | Macro-F1 | ROC-AUC | PR-AUC | F1 |
| Best Baseline | $87.90_{\pm0.00}$ | $85.28_{\pm0.00}$ | $78.36_{\pm0.16}$ | $72.57_{\pm0.48}$ | $77.93_{\pm0.56}$ |
| PA+AP+VP | $89.35_{\pm0.06}$ | $87.10_{\pm0.09}$ | $83.77_{\pm0.00}$ | $78.56_{\pm0.00}$ | $83.58_{\pm0.00}$ |
| PP+PA+PV+AP+VP | $89.32_{\pm0.07}$ | $87.01_{\pm0.13}$ | $82.92_{\pm0.00}$ | $77.67_{\pm0.00}$ | $82.63_{\pm0.01}$ |
| VP+PAP+PVP+APA | $89.41_{\pm0.04}$ | $87.18_{\pm0.07}$ | $83.14_{\pm0.00}$ | $77.95_{\pm0.00}$ | $82.84_{\pm0.00}$ |
| PP+PA+AP+VP+APA | $89.43_{\pm0.08}$ | $87.15_{\pm0.15}$ | $83.50_{\pm0.01}$ | $78.21_{\pm0.00}$ | $83.33_{\pm0.00}$ |
| PP+PA+AP+VP+PVP+APA | $89.35_{\pm0.09}$ | $87.02_{\pm0.15}$ | $82.89_{\pm0.01}$ | $77.60_{\pm0.00}$ | $82.63_{\pm0.01}$ |
| PP+PV+VP+PAP+PVP+APA | $89.36_{\pm0.05}$ | $87.04_{\pm0.05}$ | $82.91_{\pm0.00}$ | $77.62_{\pm0.01}$ | $82.65_{\pm0.01}$ |
| PP+PA+PV+AP+VP+PAP+PVP+APA | $89.17_{\pm0.13}$ | $86.91_{\pm0.11}$ | $81.04_{\pm0.00}$ | $75.78_{\pm0.01}$ | $80.43_{\pm0.02}$ |
| All meta-paths | $88.86_{\pm0.11}$ | $86.44_{\pm0.12}$ | $80.58_{\pm0.02}$ | $75.14_{\pm0.03}$ | $80.08_{\pm0.06}$ |
| k-hop-based subgraph | $89.13_{\pm0.37}$ | $86.60_{\pm0.69}$ | $83.45_{\pm0.01}$ | $78.33_{\pm0.00}$ | $83.15_{\pm0.02}$ |

Table 10: Results of neighbor sampling on the DBLP dataset.

| Meta-path | Node classification | | Link Prediction | | |
| --- | --- | --- | --- | --- | --- |
| | Micro-F1 | Macro-F1 | ROC-AUC | PR-AUC | F1 |
| Best Baseline | $87.90_{\pm0.00}$ | $85.28_{\pm0.00}$ | $78.36_{\pm0.16}$ | $72.57_{\pm0.48}$ | $77.93_{\pm0.56}$ |
| PP+PA+AP+VP+APA | $89.43_{\pm0.08}$ | $87.15_{\pm0.15}$ | $83.50_{\pm0.01}$ | $78.21_{\pm0.00}$ | $83.33_{\pm0.00}$ |
| PP+PA+AP+VP+APA(sample) | $89.42_{\pm0.08}$ | $87.14_{\pm0.00}$ | $83.43_{\pm0.01}$ | $78.11_{\pm0.00}$ | $83.27_{\pm0.01}$ |

overhead while retaining the essential structural information. Specifically, on the DBLP dataset, for the meta-path combination PP+PA+AP+VP+APA, we sample 50% of the neighbors for each meta-path to construct the subgraph summaries, denoted as PP+PA+AP+VP+APA(sample). The results are shown in Table 10. As observed, the model performance remains stable, confirming that neighbor sampling can effectively handle a large number of neighbors while preserving strong and robust performance.

**K-hop Sampling.** In scenarios where explicit meta-paths are unavailable, we can adopt a k-hop subgraph sampling strategy. For each target node, we sample a portion of its k-hop neighbors together with their connecting edges. Using the textual attributes of both (k-1)-hop and k-hop nodes and their edges, we generate separate textual summaries for each hop, which are then distilled by the PLM into graph tokens: [H1], [H2], . . . , [Hk]. The graph-aware prompt is constructed as follows: *The target node's textual information. The extracted 1-hop subgraph is summarized as: [H1]. The extracted 2-hop subgraph is summarized as: [H2] . . . The extracted k-hop subgraph is summarized as: [Hk].* Specifically, on the DBLP dataset, we sample 30% of the 3-hop neighbors along with their edges to construct the k-hop-based subgraph. We evaluate this variant under the training-free setting, and the results are reported in Table 9. As observed, the performance is comparable to that of the meta-path-based approaches, demonstrating that our framework can generalize effectively even when explicit meta-paths are not available.

### F.7 PARAMETER ANALYSIS

**Impact of Pre-defined Meta-paths.** To evaluate the effect of meta-path complexity, we compare HierPromptLM with a variant, **HierPromptLM-DMeta**, on the DBLP dataset. HierPromptLM-DMeta uses simpler meta-paths (e.g., paper–paper, paper–author, paper–venue), while HierPromptLM incorporates additional higher-order paths such as paper–author–paper, paper–venue–paper, and author–paper–author. Results are shown in Figure 6(a).

As observed, HierPromptLM outperforms HierPromptLM-DMeta on node classification, benefiting from the richer structural context provided by complex meta-paths, which enhances node representations. However, it slightly underperforms on link prediction, since simpler, direct paths (e.g., paper–paper, paper–author) are more informative for link prediction on DBLP. In this case, complex meta-paths may introduce noise or overcomplicate representations, reducing effectiveness for link prediction tasks.

**Impact of Negative Sampling Ratio.** To validate the impact of negative sampling ratio $nsr$ in the HGA-NSP task, we vary $nsr$ in $\{1, 2, 3, 4\}$ on the DBLP dataset. Results for both tasks are shown in

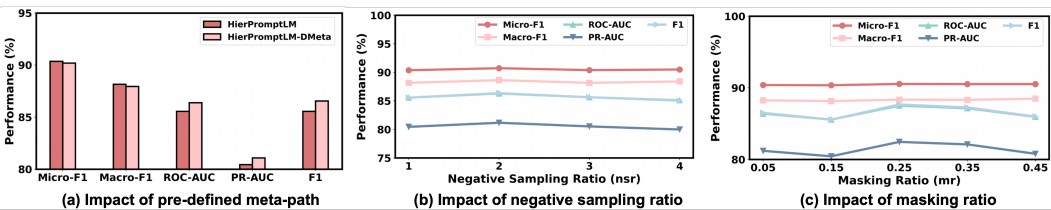

Figure 6: Parameter analysis.

Figure 6 (b). As can be observed, model performance generally improves at first and then declines, with the best results achieved at $nsr = 2$. A moderate $nsr$ introduces more informative negative samples, enhancing the model's ability to distinguish between positives and negatives. However, excessively high values can overwhelm the model, shifting its focus toward identifying negatives and hindering its ability to learn meaningful positive patterns, ultimately leading to performance degradation.

**Impact of Masking Ratio.** To evaluate the effect of the masking ratio $mr$ in the HGA-MLM task, we vary $mr$ in $\{0.05, 0.15, 0.25, 0.35, 0.45\}$ on the DBLP dataset. Results are shown in Figure 6 (c). From the results, we observe that the performance initially tends to improve and peaks at $mr = 0.25$, after which it declines. Notably, this differs from the standard Bert MLM setting, where the optimal masking ratio is typically $0.15$. This implies that when adapting Bert's standard MLM task to HTRNs, the proposed HTRN-tailored HGA-MLM task requires a higher masking ratio to better capture both textual and heterogeneous structural information on the DBLP dataset, leading to more effective representation learning on HTRNs.

## G    IMPACT STATEMENT

This paper aims to advance the field of graph learning by addressing a critical gap in representation learning on HTRNs. We propose HierPromptLM, a pure PLM-based framework that introduces two novel *HTRN-tailored pretraining* tasks and an innovative hierarchical prompt by integrating both text data and heterogeneous structures within a unified textual space, effectively overcoming limitations of existing methods. By achieving state-of-the-art performance in node classification, link prediction and graph classification tasks, HierPromptLM demonstrates potential for applications in diverse domains, such as academic network analysis and personalized recommendation systems, which offers significant societal benefits with no foreseeable negative impacts.

