# OpenReview forum: "HierPromptLM: Hierarchical Prompt Language Model for Heterogeneous Text-rich Networks"
_ICLR.cc/2026/Conference — Submitted to ICLR 2026_

### Official Review · Reviewer_3BE4 · 2025-10-24

**Soundness:** 3
**Presentation:** 3
**Contribution:** 2
**Rating:** 4
**Confidence:** 4

**Summary:**

The paper proposes a pure-PLM framework for heterogeneous text-rich networks that turns node/edge structure into hierarchical prompts: node-level “graph-aware” prompts distilled from meta-path subgraphs, and edge-level relation-aware prompts with a learnable relation token; the PLM is then tuned with two pretraining tasks, HGA-NSP and HGA-MLM. On DBLP/OAG/GoodReads, it reports consistent gains for node classification and link prediction, and shows variants across PLM backbones.

**Strengths:**

1. The work unifies textual space for structure+text via hierarchical prompts avoids explicit alignment between PLM and HGNN spaces.

2. The paper give clear ablations isolate the contribution of each component (HGA-MLM/NSP, graph tokens, node-specific text).

3. The model shows good performance against the baselines. Across three datasets; the training-free “frozen” variant already beats baselines, and fine-tuning adds more.

**Weaknesses:**

1. The approach hinges on predefined meta-paths and a hand-crafted “subgraph program function.” The paper does not quantify sensitivity to meta-path choice/coverage or propose automated discovery, which might threaten robustness and portability across domains.

2. Although graph summaries are distilled into tokens, the pipeline still requires offline extraction/summarization and concatenation of multiple prompts; token-length pressure is acknowledged only qualitatively. There’s no empirical end-to-end accounting of preprocessing time vs. baselines beyond a brief complexity line and appendix pointer.

3. The metric for link prediction results does not align with the previous papers. The works such as HGT and Heterformer use MMR and NDCG for link prediction, but this work use ROC-AUC, PR-AUC and F1, which may not truly reflect the model 's capability.

**Questions:**

1. How sensitive are results to the chosen meta-path set (coverage/length)?

2. What are wall-clock times for (i) subgraph extraction, (ii) textualization, and (iii) token distillation per node, and how do they scale with graph size and |M|?

3. What is the average token length of graph-aware and relation-aware prompts per dataset? Are there any truncation or budget allocation strategy across meta-paths?

4. Can the authors report the link prediction results with MMR and NDCG?

5. My **major concern** is that, in an era when LLM is highly developed, addressing only the text-rich author-paper network tasks may not constitute sufficient technical contribution for acceptance. Hence, I wonder whether the proposed method can be applied to other types of heterogeneous networks or more complicate/difficult tasks.

---

> ### Author Response · Authors · 2025-11-21
>
> Thanks for your valuable comments.
> > W1&Q1: Robustness to the Choice of Meta-path Set
>
> We would like to clarify that our method does not heavily rely on predefined meta-paths, and the submitted version already includes two experiments demonstrating the model’s robustness beyond manually defined meta-paths.
>
> - **Meta-path Sampling.** To assess the model’s robustness, we perform a Meta-path Sampling experiment that flexibly samples subsets of informative meta-paths. Specifically, we sample meta-paths from a larger pool: {PP, PA, PV, AP, VP, PAP, PVP, APA, PAPAP, APAPA, VPVPV}. The detailed settings and results are provided in Appendix F.6 of the submitted paper (“Meta-path Sampling”). Across all sampled combinations, our method consistently outperforms the best baseline, showing strong robustness even when only a subset of meta-paths is used.
> - **K-hop Sampling (without explicit meta-paths).** For scenarios where suitable meta-paths are unclear or unavailable, we adopt a k-hop subgraph sampling strategy. For each target node, we sample a portion of its k-hop neighbors together with their connecting edges. Using the textual attributes of both (k-1)-hop and k-hop nodes and their edges, we generate separate textual summaries for each hop. The experimental details and results are reported in Appendix F.6 of the submitted paper (“K-hop Sampling”). As observed, the performance is comparable to that of the meta-path–based approaches, demonstrating that our framework can generalize effectively even when explicit meta-paths are not available.
>
> > W2&Q2: Time Analysis
>
> For each node, the construction of the graph-aware prompt consists of three components:
> - **Meta-path subgraph extraction.** Extracting a meta-path-guided subgraph requires expanding typed neighbors, whose complexity is O((m/nt)^L), where n and m denote the number of nodes and edges, t is the number of node types, and L is the meta-path length. Thus, for |M| meta-paths, the total cost becomes O(|M|(m/nt)^L).
> - **Subgraph Textualization.** Converting the extracted subgraph into a textual summary requires traversing its nodes and edges and filling deterministic templates, giving O(|M|(n_s+m_s)), where n_s and m_s dente the number of nodes and edges in the subgraph.
> - **Graph token distillation.** Each textual summary is encoded by a frozen PLM encoder and pooled into a single graph token vector, giving the dominant cost: O(|M|TH), where T is the input length and H is the hidden size.
>
> Putting all components together, the per-node complexity is O(|M|((m/nt)^L+(n_s+m_s)+TH))≈O(|M|TH), because the PLM encoding term TH dominates in practice. All these computations are performed offline and are fully parallelizable across nodes and meta-paths. With parallel processing, the effective per-meta-path and per-node cost is reduced to O(TH). Therefore, our proposed prompt construction remains scalable even for large graphs.
>
> Furthermore, under the default meta-path set {PA, PP, PV, PAP, PVP, AP, APA, VP} used in our main results, we evaluate the empirical per-node runtime of each offline component: (i) Meta-path subgraph extraction: 0.24 ms per node for all meta-paths. (ii) Subgraph textualization: 0.15 ms per node for all meta-paths. (iii) Graph token distillation: 24.64 ms per node for all meta-paths. These measurements indicate that the total offline cost per node remains extremely small in practice, further confirming the lightweightness and scalability of our prompt construction pipeline.
>
> Our experiments already include large-scale HTRNs such as GoodReads, which contains over 20,000 nodes and 500,000 edges, further validating the practical scalability of the hierarchical prompt construction. For very large graphs, we can additionally adopt meta-path sampling and neighbor sampling strategies, where only a subset of meta-paths and neighbors is used when constructing graph-aware prompts. As shown in Appendix F.6 of our submitted paper, these sampling strategies effectively preserve model performance, demonstrating the robustness and scalability of our prompt construction under sampling.
> > W3&Q4: Metrics
>
> Thanks for your suggestions, and we will include it in a new version later.
>
> > Q3: Prompt Length
>
> We report the average token length of both graph-aware and relation-aware prompts for all datasets in the table below. As shown, the overall prompt lengths remain moderate. This is because our pipeline first employs a frozen PLM to distill compact graph tokens. Therefore, even though each node may involve multiple meta-paths, the constructed prompts only encode the meaning conveyed by each meta-path and its associated distilled graph tokens, rather than the full textual content of each subgraph.
> Method|DBLP|OAG|GoodReads
> -|-|-|-
> Graph-awre Prompts|140.31|129.95|134.05
> Relation-aware Prompts|510.64|442.64|427.73
>
> If exceptionally long prompts occur—which is rare—we apply the same truncation strategy used in baseline methods, enforcing a maximum length of 1024 tokens.

---

> > ### Author Response · Authors · 2025-11-21
> >
> > Thanks for your insightful comments.
> >
> > > Q5: Generalization
> >
> > We would like to clarify that our proposed method is designed for general heterogeneous text-rich networks (HTRNs) rather than being limited to author–paper networks. In our experiments, we evaluate the framework on three widely used and structurally diverse HTRN datasets: DBLP, OAG, and GoodReads.
> > - DBLP contains three node types (paper, author, venue) and three edge types (paper–paper citation, paper–author writing, paper–venue publication).
> > - OAG includes four node types (paper, author, field, institution) and four edge types (paper–paper citation, paper–author writing, paper–field assignment, author–institution affiliation), representing richer heterogeneous semantics.
> > - GoodReads contains three node types (book, author, publisher) and three edge types (book–book similarity/co-occurrence, book–author writing, book–publisher publication), demonstrating that our method applies beyond author–paper networks.
> >
> > In addition, we evaluate the proposed framework on multiple levels of downstream tasks, including node-level tasks (node classification), edge-level tasks (link prediction), and graph-level tasks (graph classification).
> >
> > The consistent improvements across these varied datasets and tasks demonstrate that our approach generalizes well to HTRNs with different domains, heterogeneous structures, and semantic complexities, and is not restricted to only text-rich author–paper networks.

---

### Official Review · Reviewer_jZ3q · 2025-10-24

**Soundness:** 3
**Presentation:** 3
**Contribution:** 3
**Rating:** 6
**Confidence:** 3

**Summary:**

The paper proposes HierPromptLM, a novel framework designed for representation learning on heterogeneous text-rich networks (HTRNs). These networks are composed of nodes and edges of diverse types, with textual information associated with each node. The core idea behind the paper is to integrate textual and structural information within a unified representation space without relying on separate processing or alignment of embeddings, which has been a limitation in prior work.

**Strengths:**

1. The introduction of HGA-MLM and HGA-NSP as pretraining tasks tailored to HTRNs is a strong aspect of this paper. These tasks allow the model to capture both textual and structural characteristics effectively, which could potentially be applied to other domains with heterogeneous graph data.
2. The experiments presented are thorough, demonstrating clear improvements over existing methods, especially in the case of node classification and link prediction. The paper also provides insightful comparisons with multiple baselines, including those that only leverage text or graph structures.

**Weaknesses:**

1. The process of generating hierarchical prompts and integrating them with node textual information could become computationally intensive as the size of the network increases. This scalability issue may limit the framework's applicability to massive real-world networks, where efficiency is paramount.
2. The performance of prompt-based models is known to be sensitive to the formulation of prompts. Minor changes in prompt phrasing can lead to significant variations in model outputs. This sensitivity raises concerns about the robustness of HierPromptLM, especially when deployed in dynamic environments where prompt structures might evolve or be subject to adversarial manipulations. Ensuring consistent performance despite such variations is a challenge that requires further investigation.

**Questions:**

See Weaknesses.

---

> ### Author Response · Authors · 2025-11-21
>
> Thanks for your valuable comments.
> > W1: Scalability of Prompt Construction
>
> The hierarchical prompts in our framework consist of the graph-aware prompt (per node) and the relation-aware prompt (per edge).
> 1. **Graph-aware prompt (offline):** For each node, the construction of its graph-aware prompt involves three components, as detailed below.
>
> - **(i) Meta-path subgraph extraction**
>
> Extracting a meta-path-guided subgraph requires expanding typed neighbors, whose complexity is O((m/nt)^L), where n and m denote the number of nodes and edges, t is the number of node types, and L is the meta-path length. Thus, for |M| meta-paths, the total cost becomes O(|M|(m/nt)^L).
>
> - **(ii) Subgraph Textualization**
>
> Converting the extracted subgraph into a textual summary requires traversing its nodes and edges and filling deterministic templates, giving O(|M|(n_s+m_s)), where n_s and m_s dente the number of nodes and edges in the subgraph.
>
> - **(iii) Graph token distillation**
>
> Each textual summary is encoded by a frozen PLM encoder and pooled into a single graph token vector, giving the dominant cost: O(|M|TH), where T is the input length and H is the hidden size.
>
> **Total:** Putting all components together, the per-node complexity is O(|M|((m/nt)^L+(n_s+m_s)+TH))≈O(|M|TH), because the PLM encoding term TH dominates in practice. All these computations are performed offline and are fully parallelizable across nodes and meta-paths. With parallel processing, the effective per-meta-path and per-node cost is reduced to O(TH).
>
> 2. **Relation-aware prompt:** Relation-aware prompts require only concatenating the precomputed graph-aware prompts of the two nodes with the corresponding relation token. This operation is extremely lightweight with O(1).
>
> Therefore, our proposed prompt construction remains scalable even for large graphs. Furthermore, under the default meta-path set {PA, PP, PV, PAP, PVP, AP, APA, VP} used in our main results, we evaluate the empirical per-node runtime of each offline component: (i) Meta-path subgraph extraction: 0.24 ms per node for all meta-paths. (ii) Subgraph textualization: 0.15 ms per node for all meta-paths. (iii) Graph token distillation: 24.64 ms per node for all meta-paths. These measurements indicate that the total offline cost per node remains extremely small in practice, further confirming the lightweightness and scalability of our prompt construction pipeline.
>
> Our experiments already include large-scale HTRNs such as GoodReads, which contains over 20,000 nodes and 500,000 edges, further validating the practical scalability of the hierarchical prompt construction. For very large graphs, we can additionally adopt meta-path sampling and neighbor sampling strategies, where only a subset of meta-paths and neighbors is used when constructing graph-aware prompts. As shown in Appendix F.6 of our submitted paper, these sampling strategies effectively preserve model performance, demonstrating the robustness and scalability of our prompt construction under sampling.
>
> > W2: Robustness
>
> We would like to clarify that our prompt formulation does not heavily rely on a fixed set of predefined meta-paths, and therefore is not particularly sensitive to manual meta-path choices. The submitted version already includes experiments demonstrating robustness beyond manually defined meta-paths. Specifically, we perform a Meta-path Sampling experiment that flexibly samples subsets of informative meta-paths from a large pool: {PP, PA, PV, AP, VP, PAP, PVP, APA, PAPAP, APAPA, VPVPV}.
> The detailed settings and results are provided in Appendix F.6 (“Meta-path Sampling”) of our submitted paper. Across all sampled combinations, our method consistently outperforms the best baseline, indicating strong robustness even when only partial meta-path information is available.
>
> Regarding potential dynamic or evolving environments, these scenarios lie in a different problem domain and are orthogonal to our target setting of static HTRN representation learning. Exploring robustness under such dynamic prompt conditions is an interesting future direction but beyond the scope of this work.

---

### Official Review · Reviewer_6NeT · 2025-10-25

**Soundness:** 3
**Presentation:** 3
**Contribution:** 2
**Rating:** 4
**Confidence:** 3

**Summary:**

This paper proposed a framework called HierPromptLM. A framework that combine pretrained language models like BERT and Graph structured data.

The method of this work consists of two major parts

1. The author designed meta-path based graph token distillation to extract graph information while avoid introducing too much tokens to PLMs

2. The author introduced two pretrain task called Heterogeneous Graph-aware Masked Language Modeling (HGA-MLM) and Heterogeneous Graph-aware Next Sentence Prediction (HGA-NSP) to adapt the model for graph structured data understanding.
	1. The graph structure is encoded using a summarization with a BERT model based on  meta-paths of graphs.

The baseline compared including wide variety of approach for learning heterogeneous graphs and text rich networks. The experimental results shows the approach outperformed pervious works and reached SOTA performance on downstream tasks including node classification and link prediction.

**Strengths:**

1. The paper is well written and easy to follow.
2. The figures are good and intuitive.
3. The experiment and ablation study is comprehensive.
4. The performance of the framework is good.

**Weaknesses:**

1. The approach is not really novel, the idea of pre-training on graph enhanced task, has already has been proposed in previous work like [1].

2. The motivation of graph token distillation is not clear.  As described in the paper, the approach is summarizing the meta path into text, and then encode the text into embeddings.  The reason for this is explained as " exceed PLMs’ token limits" in line 254, which can be resolved using more recent model like ModernBERT[2] or Qwen-Embedding[3] series.

3. Dataset like DBLP and OAG  are on scientific domain, domain specific backbone, like  SciBERT[4], is not adopted.

[1] Patton: Language Model Pretraining on Text-Rich Networks

[2] Smarter, Better, Faster, Longer: A Modern Bidirectional Encoder for Fast, Memory Efficient, and Long Context Finetuning and Inference

[3] Qwen3 Embedding: Advancing Text Embedding and Reranking Through Foundation Models

[4] SciBERT: A Pretrained Language Model for Scientific Text

**Questions:**

1. What is the motivation of graph tokens other than model limit ? Is it for efficiency or for performance ? If it is for efficiency, how efficient it compare with using full summaries ? If it is for performance, what is the improvement ? You may either use truncation on BERT/ALBERT, or using model that support longer context to demonstrate this.
2. Refer to other points weakness part.

---

> ### Author Response · Authors · 2025-11-21
>
> Thanks for your valuable comments.
> > W1: Novelty
>
> We clarify that our methodology makes two main contributions.
> (1) We propose the **first pure PLM-based framework** that jointly models textual information and heterogeneous structures for HTRN representation learning.
> (2) We are the **first to propose two new HTRN-tailored pretraining tasks** for PLMs for enhaning represenation learning on HTRNs.
>
> While [1] also incorporates network structure in pre-training, our work is fundamentally different in both graph setting and pre-training objectives.
> - Different graph setting. [1] is designed for homogeneous text-rich networks, where all nodes and edges share a single type. In contrast, we focus on heterogeneous text-rich networks that contain multiple node types, multiple relation types, and relation-specific semantics expressed through meta-paths. Such heterogeneous structures require effectively modeling of graph tokens and relation tokens, which [1] does not support, as it operates solely on homogeneous networks without typed nodes or typed edges.
> - Different pre-training objective. Although [1] leverages neighbor texts during pre-training, both network-contextualized MLM and masked node prediction remain link-oriented objectives, modeling only text-driven neighborhood correlations. Its objective cannot capture relation-specific semantics or meta-path structures, which are central to HTRNs. In contrast, our HTRN-tailored pre-training tasks enable the PLM to jointly learn textual information, heterogeneous structural semantics, and their interactions—capabilities that [1] does not support.
>
> > W2&Q1: Clarification on Graph Token Distillation
>
> We clarify that the motivation behind graph token distillation is to improve performance, rather than merely to address length limitations. For each target node, multiple meta-path–based subgraph summaries must be incorporated into the prompt; directly concatenating all these summaries results in excessively long and often redundant inputs. Such long prompts dilute the PLM’s attention over key structural cues, introduce additional noise, and lead to unstable representations that make reasoning more difficult. By distilling the essential information from each meta-path–based subgraph summary into a concise and semantically aligned embedding, graph token distillation enables the model to better leverage graph structures, and thus achieve more effective and consistent reasoning.
>
> To validate its effectiveness, we directly compare our distilled graph tokens with raw meta-path textual summaries (denoted as HierPromptLM-frozen-text) under the training-free setting. The comparison is conducted on the node classification task on the DBLP dataset, and the results are shown below:
> Method|Micro-F1|Macro-F1
> -|-|-
> HierPromptLM-frozen-text|86.17+-0.04|83.46+-0.12
> HierPromptLM-frozen|89.17+-0.13|86.91+-0.11
> Improvement|+3.00%|+3.45%
>
> The distilled graph token substantially outperforms using full text. This demonstrates that the distilled representation is more effective than raw textual summaries. Beyond this performance gain, graph token distillation also mitigates textual redundancy and prevents degradation caused by overly long or uneven meta-path expansions. Overall, these results confirm that graph token distillation is primarily a performance-driven design that consistently improves model effectiveness over using full textual summaries.
> > W3: PLM Backbone
>
> We would like to clarify that our work does not focus on scientific-domain language modeling, but on HTRN representation learning. Although DBLP and OAG contain scientific text, the core of our task is graph-based representation learning, rather than domain-specific text modeling. SciBERT [4] is designed for scientific NLP tasks (e.g., scientific NER) and is not intended to serve as a backbone for graph-based representation learning. Our method does not rely on scientific-domain vocabulary, linguistic priors, or domain-specific pretraining objectives. Therefore, adopting a scientific-domain PLM such as SciBERT is orthogonal to our problem and unnecessary for our framework. Notably, our focus is on building a domain-agnostic architecture that can operate on HTRNs from any domain, not only scientific datasets.
>
> We adopt Bert as the standard backbone in our main experiments to ensure fair and consistent comparison with existing baselines such as Heterformer and THLM, which are also based on Bert. This design choice allows us to isolate the performance gains brought by our framework, without introducing the confounding effects of differing PLM capacities.
>
> Further, we have experimented with various LLM backbones, including Bert, Albert, GPT2, MiniLM, T5, Distilbert, Roberta, and E5, as detailed in Section 5.3 of our submitted paper. Across all backbones, our model consistently outperforms the corresponding base models, showing the robustness and generalizability of our framework.

---

### Official Review · Reviewer_DenD · 2025-10-31

**Soundness:** 2
**Presentation:** 3
**Contribution:** 2
**Rating:** 4
**Confidence:** 3

**Summary:**

This paper presents HierPromptLM, a hierarchical prompt-based framework for learning representations on heterogeneous text-rich networks (HTRNs). It introduces graph-aware prompts to encode local structural information and relation-aware prompts to capture edge semantics, enabling unified modeling of text and graph structure within the PLM. Similar to pretraining tasks in the BERT series, the authors design HGA-MLM and HGA-NSP to improve the model's structural-textual understanding. Experiments on multiple datasets show that HierPromptLM achieves improvements over existing graph and PLM-based baselines.

**Strengths:**

1.	The paper proposes a paradigm that encodes heterogeneous structures into PLMs via interpretable textual summaries and graph tokens, avoiding the conventional two-space alignment of HGNN + PLM frameworks.
2.	The paper presents a well-documented methodology, including pseudocode, complexity analysis, and comprehensive evaluations, while maintaining competitive memory and runtime performance on large HTRNs.

**Weaknesses:**

1.	The method heavily relies on predefined meta-paths for subgraph construction and summarization. This dependency raises concerns about generalization to new datasets, especially when suitable meta-paths are unclear or unavailable. It remains unclear whether an automated and robust alternative exists when domain-specific meta-paths cannot be manually defined.
2.	The paper introduces a subgraph program function to convert sampled subgraphs into textual summaries but lacks a detailed discussion of its sensitivity — including template design, handling of long texts (truncation or selection), and noise or redundancy reduction.
3.	The process of graph token distillation from the frozen PLM is only described at a high level, missing key implementation details such as the distillation algorithm, vectorization procedure, or token dimensionality. In addition, for HTRNs with a large number of relation types, the representational capacity of a single learnable relation token may be insufficient.

**Questions:**

1.	The baseline results in the tables show several standard deviations as 0. Could you clarify if the baseline implementations used the official implementations or if multiple runs were conducted for reproducibility?
2.	Why does the frozen prompt (non-fine-tuned) outperform the fine-tuned model on certain tasks? Which types of nodes/edges benefit the most from the frozen prompt? Are there any failure cases where the frozen prompt underperforms?

---

> ### Author Response · Authors · 2025-11-21
>
> Thanks for your valuable comments.
> > W1: Generalization Beyond Predefined Meta-Paths
>
> We would like to clarify that our method does not heavily rely on predefined meta-paths, and the submitted version already includes two experiments demonstrating the model’s robustness beyond manually defined meta-paths.
> - **Meta-path Sampling.** To assess the model’s robustness, we perform a Meta-path Sampling experiment that flexibly samples subsets of informative meta-paths. Specifically, we sample meta-paths from a larger pool: {PP, PA, PV, AP, VP, PAP, PVP, APA, PAPAP, APAPA, VPVPV}. The detailed settings and results are provided in Appendix F.6 of the submitted paper (“Meta-path Sampling”). Across all sampled combinations, our method consistently outperforms the best baseline, showing strong robustness even when only a subset of meta-paths is used.
> - **K-hop Sampling (without explicit meta-paths).** For scenarios where suitable meta-paths are unclear or unavailable, we adopt a k-hop subgraph sampling strategy. For each target node, we sample a portion of its k-hop neighbors together with their connecting edges. Using the textual attributes of both (k-1)-hop and k-hop nodes and their edges, we generate separate textual summaries for each hop. The experimental details and results are reported in Appendix F.6 of the submitted paper (“K-hop Sampling”). As observed, the performance is comparable to that of the meta-path–based approaches, demonstrating that our framework can generalize effectively even when explicit meta-paths are not available.
>
> > W2: Clarification on Subgraph Program Function
> - Template design. Our work focuses on heterogeneous text-rich networks where meta-paths are naturally available. In this setting, we design a subgraph program function that converts sampled meta-path-based subgraphs into concise textual summaries. While the case where meta-paths are unavailable is not the primary setting we target, our method can naturally extend to such scenario via a k-hop subgraph sampling strategy (as described in W1). Experiments across different meta-path selections and the k-hop variant (Appendix F.6 of the submitted paper) show that the model remains robust across structural variations, indicating low sensitivity to the template design.
> - Handling of long texts. When the generated textual summary exceeds the input length limit, we follow standard practice in HTRN-based models (e.g., THLM, Heterformer) and apply maximum-length truncation. In practice, however, excessively long prompts rarely occur. This is because our pipeline first employs a frozen PLM to distill compact graph tokens; thus, even though a node may involve multiple meta-paths, the constructed prompts encode only the semantic meaning conveyed by each meta-path together with its distilled graph tokens, rather than the full textual content of the corresponding subgraphs. We report the average token length of both graph-aware and relation-aware prompts across all datasets in the table below. As shown, the overall prompt lengths remain moderate.
>
> Method|DBLP|OAG|GoodReads
> -|-|-|-
> Graph-awre Prompts|140.31|129.95|134.05
> Relation-aware Prompts|510.64|442.64|427.73
>
> - Noise or redundancy. The program function is based on meta-paths, meaning that each generated subgraph summary inherently follows a relation-specific semantic pattern. This structure naturally filters out irrelevant neighbors and prevents repetitive descriptions, thereby reducing both noise and redundancy in the produced summary.
>
> > W3.1: Clarification on Graph Token Distillation
>
> For each meta-path-based textual summary, we first feed it into a frozen PLM encoder (e.g., Bert) to obtain the contextualized token embeddings. Our distillation algorithm applies mean pooling over these token embeddings, producing a single vector that serves as the distilled graph token. This pooling step is the vectorization procedure, which converts variable-length text into a fixed-size representation. The resulting token naturally inherits the hidden size of the frozen PLM (e.g., 768 for Bert), ensuring dimensional consistency in downstream integration.
> > W3.2: Scalability of Relation Tokens
>
> We clarify that each relation type is assigned a dedicated learnable relation token, rather than sharing a single token across all relations. Each relation token is initialized and optimized during training and then stored in the vocabulary. This one-to-one design effectively captures the heterogeneity of edge types in HTRNs.
> > Q1: Clarification on Baseline Implementations
>
> We follow the official implementations for all baselines, and all baselines are run 10 times for reproducibility, as described in Appendix D of the submitted paper.

---

> > ### Author Response · Authors · 2025-11-21
> >
> > Thanks for your insightful comments.
> >
> > > Q2.1: Frozen Prompt vs. Fine-tuned Model
> >
> > Overall, the fine-tuned model achieves better performance on node classification (+1.5%) and link prediction (+5.13%), which is expected since our HGA-MLM and HGA-NSP objectives primarily enhance node- and edge-level representations, enabling the model to capture structural heterogeneity more effectively.
> >
> > For subgraph classification, the frozen prompt shows a marginal advantage (only 0.2%). Given such a small difference, we consider the two variants effectively comparable, and both substantially outperform all baselines across all downstream tasks, including node-, edge-, and graph-level evaluations.
> > > Q2.2: Benefits of Frozen Prompts
> >
> > We clarify that it is generally not feasible to determine which node types benefit the most from frozen prompts, as the commonly used HTRN benchmarks contain labels for only a single node type.
> >
> > For edge-level analysis, we provide additional measurements on the DBLP dataset. Given the learned edge embeddings, we randomly sample 1,000 testing edges for each relation type (PP, PA, PV), and compare Bert with our frozen prompt (HierPromptLM-frozen).
> > PP|ROC-AUC|PR-AUC|F1
> > -|-|-|-
> > Bert|50.26+-0.48|50.15+-0.22|35.52+-31.06
> > HierPromptLM-frozen|79.91+-0.03|74.33+-0.04|79.43+-0.02
> > Improvement|+29.65%|+24.18%|+43.91%
> >
> > PA|ROC-AUC|PR-AUC|F1
> > -|-|-|-
> > Bert|84.28+-10.16|77.49+-8.19|82.01+-21.02
> > HierPromptLM-frozen|86.70+-0.00|79.44+-0.00|87.99+-0.00
> > Improvement|+2.42%|+1.95%|+5.98%
> >
> > PV|ROC-AUC|PR-AUC|F1
> > -|-|-|-
> > Bert|94.87+-3.92|92.01+-3.70|94.74+-4.56
> > HierPromptLM-frozen|95.99+-0.03|93.03+-0.03|96.10+-0.03
> > Improvement|+1.12%|+1.02%|+1.36%
> >
> > As observed, the frozen prompt consistently outperforms Bert across all relation types, with the largest improvement occurring on PP edges.
> > > Q2.3: Clarification on Frozen Prompt Underperformance
> >
> > We further evaluate the frozen and fine-tuned variants across different relation types using the same setting as Q2.2.
> > PP|ROC-AUC|PR-AUC|F1
> > -|-|-|-
> > HierPromptLM-frozen|79.91+-0.03|74.33+-0.04|79.43+-0.02
> > HierPromptLM|81.52+-0.04|75.96+-0.04|81.28+-0.05
> > Improvement|+1.61%|+1.63%|+1.85%
> >
> > PA|ROC-AUC|PR-AUC|F1
> > -|-|-|-
> > HierPromptLM-frozen|86.70+-0.00|79.44+-0.00|87.99+-0.00
> > HierPromptLM|87.70+-0.00|80.26+-0.00|89.05+-0.00
> > Improvement|+1.00%|+0.82%|+1.06%
> >
> > PV|ROC-AUC|PR-AUC|F1
> > -|-|-|-
> > HierPromptLM-frozen|95.99+-0.03|93.03+-0.03|96.10+-0.03
> > HierPromptLM|96.41+-0.00|93.26+-0.00|96.51+-0.00
> > Improvement|+0.42%|+0.23%|+0.41%
> >
> > As shown, the fine-tuned model consistently surpasses the frozen prompt across all relation types, with the most substantial improvement observed on PP edges.

---

### Author Response · Authors · 2025-11-30
**Clarification-Related Concerns**

Dear ACs, SACs and PCs,

We sincerely appreciate the time and effort you have dedicated to reviewing our work. **In our rebuttal, we have replied each reviewer’s questions in detail.** Overall, the concerns can be grouped into two categories: clarification-related and experiment-related issues.
Due to space constraints, we first summarize the **clarification-related concerns** along with our corresponding responses below, and will provide a separate summary for the experiment-related issues thereafter.

## Generalization Beyond Predefined Meta-Paths
Addressing: W1 (Reviewer DenD), W2 (Reviewer jZ3q), W1&Q1 (Reviewer 3BE4).

We clarify that our method does not heavily rely on predefined meta-paths. **Appendix F.6 already includes two evaluations**: **(a) Meta-path Sampling**: testing diverse combinations of meta-paths for prompt construction. **(b) K-hop Sampling**: extending to scenarios where explicit meta-paths are unavailable, even though such scenarios are not our target HTRN setting. Both results show stable performance and consistent improvement over all baselines, confirming the generalization of our approach.

## Subgraph Program Function Robustness
Addressing: W2 (Reviewer DenD), Q3 (Reviewer 3BE4).

We clarify the robustness of subgraph prompgram function from three aspects below.
- **Template design**: Experiments under diverse meta-path selections (from W1 above) show stable performance, indicating low sensitivity to template variations.
- **Handling long texts**: Our additional token-length statistics show that excessively long prompts rarely occur (average graph-aware prompt length ≤ 140.31; average relation-aware prompt length ≤ 510.64), as the pipeline distills each meta-path-based subgraph summary into compact graph tokens. In the rare cases that exceed the input budget, we apply standard maximum-length truncation. (**New Experiments**)
- **Noise reduction**: The design inherently filters noise by focusing on meta-path–guided structural patterns.
## Graph Token Distillation & Relation Token
Addressing: W3 (Reviewer DenD).

We clarify the implementation details of the graph-token distillation process and the design of dedicated relation tokens.

## Baseline Implementations and Reproducibility
Addressing: Q1 (Reviewer DenD).

We highlight that **Appendix D of our submission already includes full implementations** of our method and all baselines, ensuring reproducibility.

## Novelty
Addressing: W1 (Reviewer 6NeT).

We clarify that our methodology makes two main contributions: (1) We propose the **first pure PLM-based framework** that jointly models textual information and heterogeneous structures for HTRN representation learning. (2) We are the **first to propose two new HTRN-tailored pretraining tasks** for PLMs for enhaning represenation learning on HTRNs.

Compared with [1] mentioned by reviewer 6NeT, our method targets a fundamentally different graph setting—HTRNs with multiple node and edge types—whereas [1] operates only on simple homogeneous graphs. Moreover, unlike [1]’s link-oriented pretraining objectives, our pretraining tasks explicitly capture relation-specific semantics and meta-path structural patterns, enabling the PLM to jointly model textual information, heterogeneous structural semantics, and their interactions—capabilities that [1] cannot support.

You may find more details in our rebuttal. We believe these concerns, while reasonable, are not fundamental issues, and our responses have effectively resolved them.

Thanks and Regards,

Authors

---

### Author Response · Authors · 2025-11-30
**Experiment-Related Concerns**

Dear ACs, SACs and PCs,

Thank you for your time and effort in reviewing our work. **In our rebuttal, we have replied each reviewer’s questions in detail.** Below, we summarize the **experiment-related** reviewer concerns and our responses.

## Frozen Prompt vs. Fine-tuned Model
Addressing: Q2 (Reviewer DenD).

**Overall Performance Comparison**. We clarify that the fine-tuned model outperforms the frozen version on node classification (+1.5%) and link prediction (+5.13%), with only a marginal drop (–0.2%) on subgraph classification. Importantly, both variants substantially outperform all baselines across downstream tasks. These gains stem from our prompt design together with two new pretraining tasks.

**Benefits across All Edge Types**. We further provided new experiments comparing **(1) Bert vs. frozen** (frozen outperforms Bert across all edge types, with the largest gains on PP edges: +43.91% F1, +24.18% PR-AUC, +29.65% ROC-AUC) and **(2) frozen vs. fine-tuned** (fine-tuned surpasses frozen across all edge types, with the largest gains again on PP edges: +1.85% F1, +1.63% PR-AUC, +1.61% ROC-AUC). (**New Experiments**)

## Effectiveness of Graph Token Distillation
Addressing: W2&Q1 (Reviewer 6NeT).

We clarify that the motivation is not only to address limit length, but also to improve performance. We provided additional experiments by comparing distilled graph tokens with raw meta-path textual summaries on node classification on the DBLP dataset, showing gains of up to +3.45% Macro-F1 and +3.00% Micro-F1. (**New Experiments**)

## PLM Backbone
Addressing: W3 (Reviewer 6NeT).

Our work targets on general HTRN representation learning rather than scientific-domain language modeling, so domain-specific PLMs like SciBERT (designed for scientific NLP tasks) are not included.

We further emphasize that we use Bert as the PLM backbone to ensure fairness, consistent with prior baselines.

We also highlight that Section 5.3 of our submitted paper already evaluates our framework across multiple PLM backbones (Bert, Albert, GPT2, MiniLM, T5, DistilBert, RoBerta, E5). Our method consistently outperforms each corresponding base model, demonstrating strong robustness and generalizability.


## Scalability of Prompt Construction
Addressing: W1 (Reviewer jZ3q), W2&Q2 (Reviewer 3BE4).

**Time Complexity**. We provide a detailed analysis showing that the graph-aware prompt construction (per node, offline) has complexity O(|M|TH), and the relation-aware prompt (per edge) has O(1), where |M| is the number of meta-paths, T is input length, and H is hidden size. Notably, graph-aware prompt construction can be conducted offline and fully parallelizable, reducing the effective per-meta-path and per-node cost to O(TH), ensuring scalability even for large graphs.

**Runtime**. We further report empirical per-node runtimes for graph-aware prompt construction: **(i) Meta-path subgraph extraction**: 0.24 ms, **(ii) Subgraph textualization**: 0.15 ms, and **(iii) Graph token distillation**: 24.64 ms. These results show that the total offline cost per node is extremely small. (**New Experiments**)

**Datasets.** Our experiments already include large-scale HTRNs such as GoodReads (20,000+ nodes, 500,000+ edges), further demonstrating practical scalability. For much larger graphs, meta-path sampling and neighbor sampling can be applied, and Appendix F.6 shows that these strategies preserve stable performance, underscoring the robustness and scalability of our prompt construction.

## Generalization across Diverse HTRNs
Addressing: Q5 (Reviewer 3BE4).

We clarify that our method is designed for general HTRNs, not limited to only author–paper networks. We have evaluated on three structurally diverse datasets—DBLP, OAG, and GoodReads—across various downstream tasks. The consistent improvements demonstrate strong generalization across domains, heterogeneous structures, and semantic complexities.

You may find more details in our rebuttal. We believe these concerns, while reasonable, are not fundamental issues, and our responses have effectively resolved them.

Thanks and Regards,

Authors

---

### Meta-Review · Area_Chair_k2ME · 2026-01-05

**Summary:**

Four reviewers provided detailed comments on this paper. One reviewer gave a marginally above-threshold score (6), and the other three reviewers gave marginally below-threshold scores (4, 4, 4). I have read all reviews and the rebuttal carefully. Overall, reviewers agree the writing is clear and the experimental results are strong, supported by fairly comprehensive ablations. However, the main concerns are about (1) limited novelty and unclear positioning w.r.t. prior work on PLM pretraining for text-rich networks and prompt-based HTRN modeling, (2) heavy reliance on meta-path design and a hand-crafted subgraph program function, and the robustness/portability issues brought by these design choices, and (3) incomplete benchmarking practices, especially the mismatch of link prediction metrics with common prior work (e.g., MMR/NDCG), making the comparison less convincing.

After reading the rebuttal, I think the authors addressed several clarification-type questions (e.g., providing meta-path sampling and k-hop sampling results, prompt length statistics, and a per-node runtime breakdown). However, the key concerns on novelty/positioning and the evaluation setup—especially the link prediction metric alignment—remain largely outstanding, since the authors only promise to add MMR/NDCG results in a later version and do not resolve the comparability issue in the current submission. Overall, I do not think the rebuttal is sufficient to change the overall assessment. I recommend to reject the paper.

**Reviewer Concerns:**

Reviewer DenD (rating 4) pointed out issues with hand-picked meta-paths and a hand-crafted subgraph program function, along with missing details on graph token distillation and some reporting/reproducibility concerns. The rebuttal adds robustness checks (meta-path sampling and k-hop sampling), explains the distillation step, and reports prompt lengths plus a runtime breakdown. However, the approach still depends on meta-path/template choices, and it is unclear how it will behave in other settings.

For reviewer 6NeT (rating 4), the main concerns are limited novelty, the motivation of graph token distillation beyond handling long inputs, and backbone choices. In the rebuttal, the authors explain how their setting differs from earlier homogeneous TRN work and add a small experiment supporting distillation. Still, the novelty concern remains, and the rebuttal does not really follow through on benchmarking against stronger/long-context alternatives.

Reviewer jZ3q (rating 6) mainly questioned scalability and prompt robustness. The rebuttal provides a cost analysis and per-node runtime numbers, and argues the heavy steps are offline and parallelizable. I think this concern is mostly addressed.

Reviewer 3BE4 (rating 4) raised concerns about meta-path sensitivity, preprocessing cost, prompt length, metric mismatch for link prediction, and whether it generalizes beyond author–paper graphs. The rebuttal answers most of these with more statistics and clarifications. However, the link prediction metric alignment (e.g., MMR/NDCG) is still missing in the current version (only promised later), which makes the comparison hard to judge.

**Reviewer Scores:**

No reviewers participated in the discussion. Based on the rebuttal and the original reviews, I think DenD (4), 6NeT (4), and 3BE4 (4) will likely keep their scores, since their main concerns are only partially addressed and the link prediction metric alignment is still outstanding. I think jZ3q will also keep the score (6).

---

### Decision · Program_Chairs · 2026-01-26

Reject